# Causal Effect Identifiability in the Presence of Latent Confounders Without Auxiliary Variables

Xiu-Chuan Li [1]   James Kwok [2]   Jiaxian Guo [3]   Tongliang Liu [1]

## Abstract

It is a fundamental challenge to ascertain whether the causal effect of a treatment on an outcome is identifiable in the presence of latent confounders, which serves as the logical prerequisite for recovering the causal effect in a partially observed system. While prior literature demonstrates that the causal effect is identifiable when there exist auxiliary variables subject to stringent structural constraints, this paper investigates identifiability of the causal effect without such variables. This means that we ground identifiability solely in the joint distribution of the treatment-outcome pair, which constitutes the irreducible statistical basis for causal effect identification. Focusing on linear structural causal models (SCMs), we provide a nuanced and complete characterization of identifiability of the causal effect contingent on the distributional properties of exogenous noises. Specifically, we formulate a set of mutually exclusive and collectively exhaustive conditions regarding the Gaussianity of exogenous noises, ascertain under which conditions the causal effect is identifiable and under which it is not, while also quantifying the cardinality of the feasible solution set for the unidentifiable cases. Finally, we empirically validate our theoretical findings.

## 1. Introduction

Recovering the causal effect of a treatment on an outcome is a central challenge in empirical research across disciplines such as social sciences (Sobel, 2000; Ogburn et al., 2024), public health (Hernán & Robins, 2006; Boon et al., 2021), and agriculture (Wuepper & Finger, 2023). While

Randomized Controlled Trials (RCTs) represent the gold standard, they are often prohibitively expensive or even infeasible due to ethical and legal constraints. Consequently, researchers have increasingly focused on estimating causal effects from observational (non-experimental) data. In this pursuit, they must first ascertain: is the causal effect of interest identifiable? In other words, can the causal effect be uniquely recovered from the observational population distribution? Ascertaining the identifiability status delineates the boundary between what is theoretically learnable and what remains intrinsically ambiguous, serving as the logical prerequisite for estimating the causal effect. Indeed, a problem should be established as theoretically well-posed before any empirical effort is dedicated to its solution.

In the standard setting where the causal direction is from the treatment to the outcome and there is no selection bias, the absence of confounders ensures that the observational distribution coincides with the interventional distribution, rendering the causal effect trivially identifiable. When confounders exist and are fully observed, the spurious correlations they induce can be eliminated using standard adjustment techniques like the backdoor formula (Pearl, 1995), thereby ensuring that the causal effect remains identifiable. However, in a partially observed system, the identifiability status of the causal effect becomes non-trivial. Prior literature has established various sufficient conditions that render the causal effect identifiable in the presence of latent confounders (Pearl, 1995; Angrist et al., 1996; Kuroki & Pearl, 2014). Most of these contributions rely on auxiliary variables[1], which are observed variables other than the treatment and the outcome. Moreover, these auxiliary variables must satisfy many stringent structural constraints as depicted in Figure 1. Although these works offer valuable insights, they are confined to special cases where such constraints are strictly satisfied, leaving a significant void in identifiability theory.

In this paper, we investigate causal effect identifiability in

---

[1]Sydney AI Centre, The University of Sydney [2]Department of Computer Science and Engineering, The Hong Kong University of Science and Technology [3]Google Research. Correspondence to: Tongliang Liu <tongliang.liu@sydney.edu.au>.

*Proceedings of the 43$^{rd}$ International Conference on Machine Learning*, Seoul, South Korea. PMLR 306, 2026. Copyright 2026 by the author(s).

---

[1]Although Rothenhäusler et al. (2019); Kivva et al. (2025) do not require auxiliary variables, they necessitate distributions from multiple environments subject to specific constraints (see Section 4 for more details). This paper does not consider such particular scenarios.

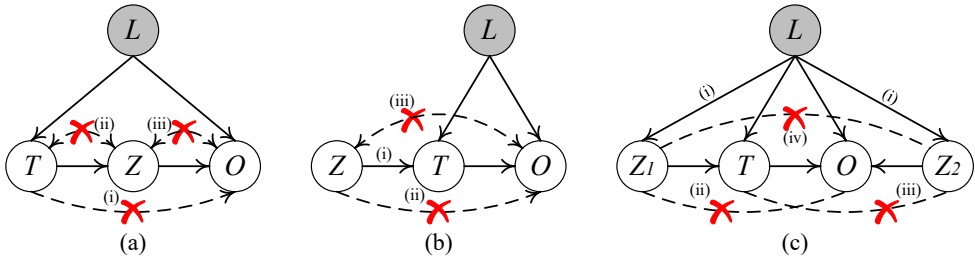

*Figure 1.* Illustration of typical auxiliary variables, where $T$ refers to the treatment, $O$ refers to the outcome, $L$ is a latent confounder.
(a): $Z$ is a valid mediator if (i) $Z$ blocks all directed paths from $T$ to $O$, (ii) there is no unblocked *backdoor path* from $T$ to $Z$ (i.e., path between $T$ and $Z$ that contains an arrow into $T$), and (iii) $T$ blocks all backdoor paths from $Z$ to $O$. (Pearl, 1995)
(b): $Z$ is a valid instrument if (i) $Z$ is a cause of $T$, (ii) $T$ blocks all directed paths from $Z$ to $O$, and (iii) there is no unblocked backdoor path from $Z$ to $O$. (Angrist et al., 1996)
(c): $(Z_1, Z_2)$ is a pair of valid proxies if (i) $L$ is a common cause of $Z_1$ and $Z_2$, (ii) $Z_1 \perp\!\!\!\perp O|\{L, T\}$, which means that there is no extra *trek* between $Z_1$ and $O$ (i.e., path between $Z$ and $O$ that contains no collider), (iii) $Z_2 \perp\!\!\!\perp T|L$, which means that there is no extra trek between $Z_2$ and $T$, and (iv) $Z_1 \perp\!\!\!\perp Z_2|L$, which means that there is no extra trek between $Z_1$ and $Z_2$. (Kuroki & Pearl, 2014)

the presence of latent confounders without auxiliary variables. In this setting, we say the causal effect is identifiable if and only if it can be uniquely determined from only the joint distribution of the treatment-outcome pair, to the exclusion of any other variables. Focusing on linear structural causal models (SCMs), which enjoy widespread popularity in the causal inference literature (Salehkaleybar et al., 2020; Rothenhäusler et al., 2021; Kivva et al., 2023; Burauel, 2023; Xie et al., 2024a; Tramontano et al., 2025), and letting $T$ denote the treatment, $O$ denote the outcome, and $L_1, ..., L_n$ denote the latent confounders, we characterize identifiability of the causal effect contingent upon the distributional properties of exogenous noises as follows:

1. Suppose the exogenous noise of $T$ is non-Gaussian and none of the exogenous noises of $L_1, ..., L_n$ is non-Gaussian. Then, the causal effect is identifiable.

2. Suppose the exogenous noise of $T$ is Gaussian and none of the exogenous noises of $L_1, ..., L_n$ is Gaussian-separable (see Definition 3 in Section 3.1). Then, the causal effect is identifiable.

3. Suppose the exogenous noise of $T$ is non-Gaussian and at least one of the exogenous noises of $L_1, ..., L_n$ is non-Gaussian. Then, the causal effect is unidentifiable and the number of feasible solutions is exactly equal to the number of non-Gaussian exogenous noises.

4. Suppose the exogenous noise of $T$ is Gaussian and at least one of the exogenous noises of $L_1, ..., L_n$ is Gaussian-separable. Then, the causal effect is unidentifiable and can take infinitely many values.

In summary, the causal effect is identifiable if and only if the exogenous noise of $T$ sharply contrasts with those of $L_1, ..., L_n$ in Gaussianity. Notably, our characterization established above is nuanced, as it extends beyond merely ascertaining the identifiability status to quantifying the cardinality of the feasible solution set, and complete, given

that the conditions regarding the Gaussianity of exogenous noises in these results are mutually exclusive and collectively exhaustive. By grounding identifiability solely in the joint distribution of the treatment-outcome pair, which constitutes the irreducible statistical basis for causal effect identification, our results establish a fundamental theoretical baseline. This baseline dictates exactly when auxiliary variables are necessary for identifiability versus when they are theoretically redundant. To the best of our knowledge, there exist no similar results in the previous literature. Finally, we also empirically validate our theoretical findings.

## 2. Preliminary

### 2.1. Statistical Foundations

Our analysis entails two fundamental statistical concepts: the characteristic function and the cumulant. We provide their definitions and relevant properties below.

### § Cumulant

Cumulants and moments represent two informationally equivalent characterizations of a probability distribution (Stuart & Ord, 2010).

**Definition 1.** (Cumulant) Given $k$ random variables $X_1, ..., X_k$ (not necessarily distinct), the cumulant $\mathrm{cum}(X_1, ..., X_k)$ is defined as

$$\mathrm{cum}(X_1, ..., X_k)$$
$$= \sum_{(A_1,...,A_l)} (-1)^{l-1}(l-1)! \prod_{i=1}^{l} \mathbb{E}\left[\prod_{j \in A_i} X_j\right], \quad (1)$$

where the sum is taken over all partitions $(A_1, ..., A_l)$ of $\{1, ..., k\}$.

In the following, for ease of exposition, we denote

$\mathrm{cum}(\underbrace{X, \ldots,}_{k_1 \text{ times}} \underbrace{Y, \ldots}_{k_2 \text{ times}})$ and $\mathrm{cum}(\underbrace{X, \ldots}_{k \text{ times}})$ by $\mathrm{cum}_{k_1,k_2}(X, Y)$ and $\mathrm{cum}_k(X)$ respectively. In this notation, the sum of the subscripts corresponds to the order of the cumulant. Notably, low-order cumulants reduce to familiar statistics: $\mathrm{cum}_1(X)$, $\mathrm{cum}_2(X)$, and $\mathrm{cum}_{1,1}(X, Y)$ correspond to $\mathbb{E}[X]$, $\mathrm{var}(X)$, and $\mathrm{cov}(X, Y)$, respectively. The cumulant is known to manifest the following properties (Belkin et al., 2013; Cai et al., 2023):

1. $\mathrm{cum}(X_1, ..., X_i, ..., X_j, ..., X_k) = \mathrm{cum}(X_1, ..., X_j, ..., X_i, ..., X_k)$.

2. If $\lambda$ is a constant, then $\mathrm{cum}(\lambda X_1, ..., X_k) = \lambda\mathrm{cum}(X_1, ..., X_k)$.

3. If $(X_1, ..., X_k) \perp\!\!\!\perp (Y_1, ..., Y_k)$, then $\mathrm{cum}(X_1+Y_1, ..., X_k+Y_k) = \mathrm{cum}(X_1, ..., X_k) + \mathrm{cum}(Y_1, ..., Y_k)$.

4. If $X$ is Gaussian, for any $k > 2$, $\mathrm{cum}_k(X) = 0$; otherwise, there exist infinitely many $k$ s.t. $\mathrm{cum}_k(X) \neq 0$.

### § Characteristic Function

The characteristic function of any real-valued random vector completely defines its probability distribution (Stuart & Ord, 2010). In other words, two random vectors $\mathbf{X}$ and $\mathbf{Y}$ are identically distributed if and only if their characteristic functions are identical.

**Definition 2.** (Characteristic function) Given a random vector[2] $\mathbf{X} = (X_1, ..., X_k)$, its characteristic function $\phi_{\mathbf{X}}(\mathbf{t})$ is defined as

$$\phi_{\mathbf{X}}(\mathbf{t}) = \mathbb{E}[\exp(j\mathbf{t}\mathbf{X}^T)], \qquad (2)$$

where $\mathbf{t} = (t_1, ..., t_k)$ is the argument and $j^2 = -1$.

From Equation (2), we can easily derive that

1. Given a random vector $\mathbf{X}$ and a constant matrix $A$, $\phi_{\mathbf{X}A}(\mathbf{t}) = \phi_{\mathbf{X}}(\mathbf{t}A^T)$.

2. Given $k$ independent random vectors $\mathbf{X}_1, ..., \mathbf{X}_k$, $\phi_{(\mathbf{X}_1,...,\mathbf{X}_k)}((\mathbf{t}_1, ..., \mathbf{t}_k)) = \prod_{i=1}^{k} \phi_{\mathbf{X}_i}(\mathbf{t}_i)$.

3. Given $k$ independent random vectors $\mathbf{X}_1, ..., \mathbf{X}_k$, $\phi_{\sum_{i=1}^{k} \mathbf{X}_i}(\mathbf{t}) = \prod_{i=1}^{k} \phi_{\mathbf{X}_i}(\mathbf{t})$.

4. Given a random vector $\mathbf{X} \sim \mathcal{N}(\mu, \Sigma)$, $\phi_{\mathbf{X}}(\mathbf{t}) = \exp\left(j\mu\mathbf{t}^T - \frac{1}{2}\mathbf{t}\Sigma\mathbf{t}^T\right)$.

### 2.2. Problem Setting

We consider a structural causal model (Pearl et al., 2016) (SCM) $\mathcal{M}$ whose causal structure is shown as Figure 2, in which the causal effect of $T$ on $O$ is defined as:

$$\mathbb{E}[O|do(T = 1)] - \mathbb{E}[O|do(T = 0)], \qquad (3)$$

---

[2]Unless otherwise specified, all vectors in this paper are assumed to be row vectors.

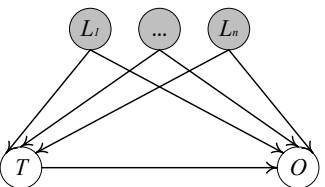

*Figure 2.* A causal structure with latent confounders and no auxiliary variables, where $T$ represents the treatment, $O$ represents the outcome, and $L_1, .., L_n$ are latent confounders.

where $do(T = 1)$ denotes severing the causal edges from $L_1, ..., L_n$ to $T$ and forcing $T$ to take the value 1 (Pearl et al., 2016). By leveraging linearization techniques around the operating point of a system, the structural functions can often be approximated by linear functions, resulting in a linear SCM (Rosenström et al., 2012; Moneta et al., 2013; Campomanes et al., 2014; Ogawa et al., 2022; Moriyama & Kuwano, 2022; Kurotani et al., 2024). Moreover, a significant body of very recent causal inference research (Li et al., 2025; Sanjaroonpouri & Ramazi, 2025; Oh et al., 2025; Dai et al., 2025; Tramontano et al., 2025; Kivva et al., 2025; Briefs & Bläser, 2025) has also centered their analysis on linear SCMs. Within a linear SCM, $L_i, T, O$ can be expressed as:

$$\begin{aligned} L_i &= \epsilon_{L_i}, \ 1 \leq i \leq n, \\ T &= \sum_{i=1}^{n} L_i + \epsilon_T, \\ O &= \alpha T + \sum_{i=1}^{n} \beta_i L_i + \epsilon_O, \end{aligned} \qquad (4)$$

where $\alpha$ is the direct causal strength from $T$ to $O$ and $\beta_i$ is the direct causal strength from $L_i$ to $O$; all exogenous noises $\epsilon_{L_1}, ..., \epsilon_{L_n}, \epsilon_T, \epsilon_O$ are non-constant and jointly independent, which are assumed to have a mean of zero without loss of generality. Within such a linear SCM, Equation (3) equals the $\alpha$. Based on Equation (4), $L_i, T, O$ can all be written as linear combinations of exogenous noises:

$$\begin{aligned} L_i &= \epsilon_{L_i}, \ 1 \leq i \leq n, \\ T &= \sum_{i=1}^{n} \epsilon_{L_i} + \epsilon_T, \\ O &= \sum_{i=1}^{n} (\alpha + \beta_i)\epsilon_{L_i} + \alpha\epsilon_T + \epsilon_O. \end{aligned} \qquad (5)$$

Throughout this paper, besides linear SCMs, we also make a parametric assumption that does not compromise generality and a distributional assumption following Kivva et al. (2023; 2025) below.

**Assumption 1.** (i) For any $i$, $\beta_i \neq 0$. (ii) For any $i \neq j$, $\beta_i \neq \beta_j$.

> **Remark.** Assumption 1 does not compromise generality. Specifically, if $\beta_i = 0$, $L_i$ can be absorbed into $T$ by changing $\epsilon_T$ to $\epsilon_T + \epsilon_{L_i}$; if $\beta_i = \beta_j$, $L_i$ and $L_j$ can be merged into a single latent confounder $L_i + L_j$. Particularly, this assumption allows that $\alpha = 0$ or $\alpha + \beta_i = 0$.

**Assumption 2.** For each $\epsilon \in \{\epsilon_{L_1}, ..., \epsilon_{L_n}, \epsilon_T, \epsilon_O\}$ and $k > 0$, $|\mathrm{cum}_k(\epsilon)| < \infty$.

**Remark.** Essentially, Assumption 2 states that $\epsilon$ has finite cumulants of all orders. Based on Equation (1), it is evident that any distribution with a bounded support satisfies this assumption. In addition, many common distributions with an unbounded support such as the Gaussian distribution, exponential distribution, and Laplace distribution also satisfy this assumption. Notably, Kivva et al. (2023; 2025) explicitly assume that $\epsilon$ has finite moments of all orders, which is equivalent to this assumption. Also, Chen et al. (2024); Tramontano et al. (2025) make this assumption implicitly.

In fact, this assumption is not strictly necessary. For instance, we can replace it with a weaker one that there exists $K \geq 2n + 2$ s.t. for each $\epsilon \in \{\epsilon_{L_1}, ..., \epsilon_{L_n}, \epsilon_T, \epsilon_O\}$, $|\text{cum}_K(\epsilon)| < \infty$ and if $\epsilon$ is non-Gaussian, there exists $2n + 2 \leq k_\epsilon \leq K$ s.t. $\text{cum}_{k_\epsilon}(\epsilon) \neq 0$. Under this assumption, we only consider cumulants of order up to $K$, and it is evident that all subsequent derivations involving cumulants remain valid when restricted to these orders.

In particular, we do not assume that the exact value of $n$ is known a priori.

## 3. Identifiability

### 3.1. Theoretical Results

First, writing $X \overset{d}{=} Y$ to denote that the random variables $X$ and $Y$ are identically distributed, we present the definition of Gaussian-separable and provide some relevant lemmas.

**Definition 3.** (Gaussian-separable) Given a random variable $X$, we say $X$ is Gaussian-separable if there exist $X_1, X_2$ s.t. (i) $X \overset{d}{=} X_1 + X_2$, (ii) $X_1 \perp\!\!\!\perp X_2$, and (iii) $X_2$ is Gaussian with zero mean. Notably, $X_1$ is allowed to be degenerate.

**Remark.** Naturally, any Gaussian distribution is Gaussian-separable. In addition, a non-Gaussian distribution is also Gaussian-separable if it is the sum of several distributions, of which at least one is Gaussian. A typical example is the ex-Gaussian distribution widely-used in psychology (Matzke & Wagenmakers, 2009), which is a sum of a Gaussian distribution and an exponential distribution.

Conversely, most canonical non-Gaussian distributions are not Gaussian-separable. Trivially, any distribution with a bounded (e.g., the uniform distribution) or a semi-bounded support (e.g., the exponential distribution) is not Gaussian-separable. In addition, some distributions with an unbounded support such as the Laplace distribution are also not Gaussian-separable. Notably, "not Gaussian-separable" is equivalent to "without Gaussian component" in Matias (2002).

It is well-known that the sum of several independent Gaussian variables is Gaussian. Also, by Cramér's decomposition theorem (Cramér, 1936), if the sum of several independent variables is Gaussian, then each summand is also Gaussian. Based on this, we derive the following lemma and corollary.

**Lemma 1.** *If $X$ is Gaussian-separable, then there exist $X_1$ and $X_2$ s.t. (i) $X \overset{d}{=} X_1 + X_2$, (ii) $X_1 \perp\!\!\!\perp X_2$, (iii) $X_1$ is not Gaussian-separable, and (iv) $X_2$ is Gaussian with zero mean. Moreover, the distributions of $X_1$ and $X_2$ are unique.*

**Intuition.** Lemma 1 shows that a Gaussian-separable distribution can be decomposed into the sum of a distribution that is not Gaussian-separable and a Gaussian distribution. Moreover, this decomposition is unique.

Intuitively, $X_1$ represents the irreducible non-Gaussian component of $X$ from which no further additive Gaussian term can be extracted, while $X_2$ corresponds to the maximal Gaussian component of $X$. For instance, if $X$ is Gaussian with zero mean, then $X_1$ degenerates to zero while $X_2$ is identical to $X$ in distribution.

**Corollary 1.** *If $X$ is Gaussian-separable, then there exist $X_1'$ and $X_2'$ s.t. (i) $X \overset{d}{=} X_1' + X_2'$, (ii) $X_1' \perp\!\!\!\perp X_2'$, (iii) $X_1'$ is Gaussian-separable, and (iv) $X_2'$ is Gaussian with zero mean. Moreover, the distributions of $X_1'$ and $X_2'$ can take infinitely many forms.*

**Intuition.** Corollary 1 shows that a Gaussian-separable distribution can also be decomposed into the sum of a Gaussian-separable distribution and a Gaussian distribution. Moreover, unlike the uniqueness in Lemma 1, there exist infinitely many such decompositions.

Intuitively, we can allocate an arbitrary fraction of the maximal Gaussian component of the original distribution to its irreducible non-Gaussian component, yielding such a decomposition. For instance, if $X$ is Gaussian with zero mean, then $X_1'$ and $X_2'$ can be arbitrary zero-mean Gaussian variables provided that $\text{var}(X_1') + \text{var}(X_2') = \text{var}(X')$ and $\text{var}(X_1')\text{var}(X_2') \neq 0$.

Equipped with the concept of Gaussian-separability, we formulate four mutually exclusive and collectively exhaustive conditions on the Gaussianity of exogenous noises and present corresponding theorems characterizing identifiability of the causal effect of $T$ on $O$, or more specifically, ascertaining whether the $\alpha$ in Equation (4) can be uniquely determined by the joint distribution of $(T, O)$.

§ **Condition 1: Identifiable**

**Condition 1.** $\epsilon_T$ is non-Gaussian and for any $\epsilon \in \{\epsilon_{L_1}, ..., \epsilon_{L_n}\}$, $\epsilon$ is Gaussian.

**Theorem 1.** *Suppose Condition 1 holds. Then, the causal effect of $T$ on $O$ is identifiable.*

**Intuition.** According to Equation (5), $\epsilon_T$ enters $T$ with a scaling factor of 1 while it enters $O$ with a scaling factor of $\alpha$. Condition 1 guarantees that $\epsilon_T$ exhibits a sharp distributional contrast with $\epsilon_{L_1}, ..., \epsilon_{L_n}$. This ensures that $\epsilon_T$ leaves a distinguishable footprint in the joint distribution of $T$ and $O$. Therefore, $\alpha$ is identifiable.

First, we introduce the following Lemma 2.

**Lemma 2.** *For any integers $i > 0$ and $j \geq 0$,*

$$\text{cum}_{i,j}(T, O) = \sum_{k=1}^{n} (\alpha + \beta_k)^j \text{cum}_{i+j}(\epsilon_{L_k}) + \alpha^j \text{cum}_{i+j}(\epsilon_T).$$

Next, we leverage Lemma 2 to prove Theorem 1.

*Proof.* We prove that for any two SCMs $\mathcal{M}$ and $\tilde{\mathcal{M}}$ satisfying Condition 1 s.t. $(T, O) \overset{d}{=} (\tilde{T}, \tilde{O})$, $\tilde{\alpha} = \alpha$.

For the SCM $\mathcal{M}$, since $\epsilon_{L_1}, ..., \epsilon_{L_n}$ are all Gaussian, according to Lemma 2, for any integers $i > 0$ and $j \geq 0$ s.t. $i + j > 2$,

$$\text{cum}_{i,j}(T, O) = \alpha^j \text{cum}_{i+j}(\epsilon_T). \tag{6}$$

As $\epsilon_T$ is non-Gaussian, there exists $k > 2$ s.t. $\text{cum}_{k,0}(T, O) = \text{cum}_k(\epsilon_T) \neq 0$, so

$$\alpha = \frac{\text{cum}_{k-1,1}(T, O)}{\text{cum}_{k,0}(T, O)}.$$

For the SCM $\tilde{\mathcal{M}}$, since $(T, O) \overset{d}{=} (\tilde{T}, \tilde{O})$, for any integers $i$ and $j$, $\text{cum}_{i,j}(\tilde{T}, \tilde{O}) = \text{cum}_{i,j}(T, O)$. Similarly to Equation (6), for any integers $i > 0$ and $j \geq 0$ s.t. $i + j > 2$,

$$\text{cum}_{i,j}(\tilde{T}, \tilde{O}) = \tilde{\alpha}^j \text{cum}_{i+j}(\epsilon_{\tilde{T}}),$$

so

$$\alpha = \frac{\text{cum}_{k-1,1}(T, O)}{\text{cum}_{k,0}(T, O)} = \frac{\text{cum}_{k-1,1}(\tilde{T}, \tilde{O})}{\text{cum}_{k,0}(\tilde{T}, \tilde{O})} = \tilde{\alpha}, \tag{7}$$

this completes the proof. $\qquad\square$

## § Condition 2: Identifiable

**Condition 2.** $\epsilon_T$ is Gaussian and for any $\epsilon \in \{\epsilon_{L_1}, ..., \epsilon_{L_n}\}$, $\epsilon$ is not Gaussian-separable.

**Theorem 2.** *Suppose Condition 2 holds. Then, the causal effect of $T$ on $O$ is identifiable.*

**Intuition.** Similar to Condition 1, Condition 2 also guarantees that $\epsilon_T$ exhibits a sharp distributional contrast with $\epsilon_{L_1}, ..., \epsilon_{L_n}$. Therefore, $\alpha$ is identifiable.

*Proof.* We prove that for any two SCMs $\mathcal{M}$ and $\tilde{\mathcal{M}}$ satisfying Condition 2 s.t. $(T, O) \overset{d}{=} (\tilde{T}, \tilde{O})$, $\tilde{\alpha} = \alpha$.

Since $(T, O) \overset{d}{=} (\tilde{T}, \tilde{O})$,

$$\phi_{(T,O)}((t_1, t_2)) = \phi_{(\tilde{T}, \tilde{O})}((t_1, t_2)). \tag{8}$$

If $\epsilon_O$ is Gaussian-separable, based on Lemma 1, there exist $\epsilon_O'$ and $\epsilon_O''$ s.t.

$$\epsilon_O \overset{d}{=} \epsilon_O' + \epsilon_O'',$$
$$\epsilon_O' \text{ is not Gaussian-separable,}$$
$$\epsilon_O'' \text{ is Gaussian with zero mean,}$$
$$\epsilon_T \perp\!\!\!\perp \epsilon_{L_1} \perp\!\!\!\perp ... \perp\!\!\!\perp \epsilon_{L_n} \perp\!\!\!\perp \epsilon_O' \perp\!\!\!\perp \epsilon_O'';$$

otherwise, if $\epsilon_O$ is not Gaussian-separable, we let $\epsilon_O' = \epsilon_O$ and $\epsilon_O'' = 0$. Likewise, we construct $\epsilon_{\tilde{O}}'$ and $\epsilon_{\tilde{O}}''$ for $\epsilon_{\tilde{O}}$. Because $T$ and $O$ can be expressed as linear combinations of exogenous noises that are mutually independent (see Equation (5)), Equation (8) can be equivalently written as

$$\phi_{(\sum_{i=1}^{n} \epsilon_{L_i}, \sum_{i=1}^{n}(\alpha+\beta_i)\epsilon_{L_i}+\epsilon_O')}((t_1, t_2))\phi_{(\epsilon_T, \alpha\epsilon_T+\epsilon_O'')}((t_1, t_2))$$
$$=\phi_{(\sum_{i=1}^{n} \epsilon_{\tilde{L}_i}, \sum_{i=1}^{n}(\tilde{\alpha}+\tilde{\beta}_i)\epsilon_{\tilde{L}_i}+\epsilon_{\tilde{O}}')}((t_1, t_2))\phi_{(\epsilon_{\tilde{T}}, \tilde{\alpha}\epsilon_{\tilde{T}}+\epsilon_{\tilde{O}}'')}((t_1, t_2)). \tag{9}$$

Because $\epsilon_T$ and $\epsilon_{\tilde{T}}$ are Gaussian, each of $\epsilon_O''$ and $\epsilon_{\tilde{O}}''$ is either Gaussian or 0, and none of $\epsilon_{L_1}, ..., \epsilon_{L_n}, \epsilon_O', \epsilon_{\tilde{L}_1}, ..., \epsilon_{\tilde{L}_n}, \epsilon_{\tilde{O}}'$ is Gaussian-separable, we have

$$\phi_{(\epsilon_T, \alpha\epsilon_T+\epsilon_O'')}((t_1, t_2)) = \phi_{(\epsilon_{\tilde{T}}, \tilde{\alpha}\epsilon_{\tilde{T}}+\epsilon_{\tilde{O}}'')}((t_1, t_2)),$$

so

$$\alpha = \frac{\text{cov}(\epsilon_T, \alpha\epsilon_T + \epsilon_O'')}{\text{var}(\epsilon_T)} = \frac{\text{cov}(\epsilon_{\tilde{T}}, \tilde{\alpha}\epsilon_{\tilde{T}} + \epsilon_{\tilde{O}}'')}{\text{var}(\epsilon_{\tilde{T}})} = \tilde{\alpha}.$$

This completes the proof. $\qquad\square$

**Remark.** Since the causal effect is already identifiable, it remains trivially identifiable if additional assumptions are imposed. In particular, further assuming $n \leq 2$ and Gaussian $\epsilon_O$ preserves identifiability. This aligns with the adaptation of Corollary 2 in Sanjaroonpouri & Ramazi (2025) to our problem setting.

## § Condition 3: Unidentifiable

**Condition 3.** $\epsilon_T$ is non-Gaussian and there exists $\epsilon \in \{\epsilon_{L_1}, ..., \epsilon_{L_n}\}$ s.t. $\epsilon$ is non-Gaussian.

Denote by $\mathbb{S}$ the feasible solution set for the causal effect of $T$ on $O$, the following Theorem 3 holds.

**Theorem 3.** *Suppose Condition 3 holds. Then, the causal effect of $T$ on $O$ is unidentifiable and $|\mathbb{S}| = 1 + |\{\epsilon_{L_i}|\epsilon_{L_i} \text{ is non-Gaussian}\}|$. More specifically,*

$$\mathbb{S} = \{\alpha + \beta_i | \epsilon_{L_i} \text{ is non-Gaussian}\} \cup \{\alpha\}. \tag{10}$$

**Intuition.** Under Condition 3, the role of the non-Gaussian $\epsilon_T$ is interchangeable with that of any non-Gaussian $\epsilon_{L_i}$, hence $\alpha$, the scaling factor of $\epsilon_T$ in $O$, is not distinguishable from $\alpha + \beta_i$, the scaling factor of $\epsilon_{L_i}$.

First, we introduce a matrix $A(X, Y; p, q, r) \in \mathbb{R}^{p \times p}$ composed of cumulants of $X$ and $Y$ where $p, q, r$ can be arbitrary integers subject to

$$p \geq 1, \ q \geq 0, \ r \geq 2p + q - 1.$$

The $(i, j)$-th element of $A(X, Y; p, q, r)$ is given by

$$A(X, Y; p, q, r)_{i,j} = \mathrm{cum}_{r-(q+i+j-2), q+i+j-2}(X, Y).$$

The following Lemma 3 connects the rank of $A(T, O; p, q, r)$ to $\alpha$.

**Lemma 3.** *Let $m$ denote $|\{\epsilon_{L_i} | \epsilon_{L_i}$ is non-Gaussian$\}|$ and $m'$ denote $\mathbb{1}(\epsilon_T$ is non-Gaussian$)$ where $\mathbb{1}(\cdot)$ is the indicator function. Then, for any $p \geq \max(m + m', 1)$ and $r \geq \max(2p, 3)$, the rank of $A(T, O; p, 1, r) - \lambda A(T, O; p, 0, r)$ is less than the rank of $A(T, O; p, 0, r)$ if and only if*

$$\lambda \in \begin{cases} \{\alpha + \beta_i | \mathrm{cum}_r(\epsilon_{L_i}) \neq 0\} & \text{if } \mathrm{cum}_r(\epsilon_T) = 0 \\ \{\alpha + \beta_i | \mathrm{cum}_r(\epsilon_{L_i}) \neq 0\} \cup \{\alpha\} & \text{if } \mathrm{cum}_r(\epsilon_T) \neq 0 \end{cases}.$$

Next, we leverage Lemma 3 to prove Theorem 3.

*Proof.* We prove that the LHS of Equation (10) is both a subset and a superset of the RHS of Equation (10).

Part 1: $\mathbb{S} \supseteq \{\alpha + \beta_i | \epsilon_{L_i}$ is non-Gaussian$\} \cup \{\alpha\}$.

Because it is trivial that $\alpha \in \mathbb{S}$, it is sufficient to prove that given a SCM $\mathcal{M}$ satisfying Condition 3, if $\epsilon_{L_i}$ is non-Gaussian, then there exists a SCM $\tilde{\mathcal{M}}$ also satisfying Condition 3 s.t. $(T, O) \overset{d}{=} (\tilde{T}, \tilde{O})$ and $\tilde{\alpha} = \alpha + \beta_i$.

Suppose $\epsilon_{L_1}$ is non-Gaussian without loss of generality. Consider a SCM $\tilde{\mathcal{M}}$ in which the number of latent confounders $\tilde{n} = n$:

$$\begin{aligned} \tilde{L}_1 &= \epsilon_T, \\ \tilde{L}_i &= \epsilon_{L_i}, \ 2 \leq i \leq n, \\ \tilde{T} &= \sum_{i=1}^n \tilde{L}_i + \epsilon_{L_1}, \\ \tilde{O} &= (\alpha + \beta_1)\tilde{T} - \beta_1 \tilde{L}_1 + \sum_{i=2}^n (\beta_i - \beta_1)\tilde{L}_i + \epsilon_O. \end{aligned}$$

Writing $T, O, \tilde{T}, \tilde{O}$ as linear combinations of exogenous noises, it is trivial that $(T, O) \overset{d}{=} (\tilde{T}, \tilde{O})$. Also, it is trivial that

$$\tilde{\alpha} = \alpha + \beta_1 \in \mathbb{S}.$$

Extending this analysis to each non-Gaussian $\epsilon_{L_i}$, we can arrive at the final conclusion.

Part 2: $\mathbb{S} \subseteq \{\alpha + \beta_i | \epsilon_{L_i}$ is non-Gaussian$\} \cup \{\alpha\}$.

We prove that for any two SCMs $\mathcal{M}$ and $\tilde{\mathcal{M}}$ satisfying Condition 3 s.t. $(T, O) \overset{d}{=} (\tilde{T}, \tilde{O})$, $\tilde{\alpha} \in \{\alpha + \beta_i | \epsilon_{L_i}$ is non-Gaussian$\} \cup \{\alpha\}$.

Suppose there exists a SCM $\tilde{\mathcal{M}}$ s.t. $(\tilde{T}, \tilde{O}) \overset{d}{=} (T, O)$ and $\tilde{\alpha} \notin \{\alpha + \beta_i | \epsilon_{L_i}$ is non-Gaussian$\} \cup \{\alpha\}$. Let $m$ denote $|\{\epsilon_{L_i} | \epsilon_{L_i}$ is non-Gaussian$\}|$. According to Corollary 2 in App. A, there is $|\{\epsilon_{\tilde{L}_i} | \epsilon_{\tilde{L}_i}$ is non-Gaussian$\}| = m$. Because $\epsilon_{\tilde{T}}$ is non-Gaussian, there exists $r \geq 2m+2$ s.t. $\mathrm{cum}_r(\epsilon_{\tilde{T}}) \neq 0$. According to Lemma 3,

$$\mathrm{rk}(A(\tilde{T}, \tilde{O}; m+1, 1, r) - \tilde{\alpha}A(\tilde{T}, \tilde{O}; m+1, 0, r))$$
$$< \mathrm{rk}(A(\tilde{T}, \tilde{O}; m+1, 0, r)).$$

Because $(\tilde{T}, \tilde{O}) \overset{d}{=} (T, O)$,

$$\mathrm{rk}(A(T, O; m+1, 1, r) - \tilde{\alpha}A(T, O; m+1, 0, r))$$
$$< \mathrm{rk}(A(T, O; m+1, 0, r)).$$

According to Lemma 3, we have

$$\tilde{\alpha} \in \{\alpha + \beta_i : \mathrm{cum}_r(\epsilon_{L_i}) \neq 0\} \cup \{\alpha\},$$

which leads to a contradiction. □

**Remark.** If we assume that $\epsilon_{L_1}, ..., \epsilon_{L_n}, \epsilon_T, \epsilon_O$ are all non-Gaussian, the linear SCM reduces to a latent variable linear non-Gaussian acyclic model (lvLiNGAM) (Hoyer et al., 2008). Notably, Kivva et al. (2023); Tramontano et al. (2024; 2025) all focused on causal effect identification in lvLiNGAM. In particular, Theorem 3.1 in Tramontano et al. (2025) discusses the scenario without auxiliary variables. By additionally assuming that non-Gaussian variables have non-zero cumulants of all orders, their Theorem 3.1 implies that the causal effect has at most $n + 1$ possible values.

It is trivial that under the assumptions of lvLiNGAM and non-zero cumulants, our proof remains valid, which suggests that $|\mathbb{S}| = n + 1$ in this case. In other words, our result tightens the upper bound of at most $n + 1$ in Tramontano et al. (2025)'s Theorem 3.1 to precisely $n + 1$.

## § Condition 4: Unidentifiable

**Condition 4.** $\epsilon_T$ is Gaussian and there exists $\epsilon \in \{\epsilon_{L_1}, ..., \epsilon_{L_n}\}$ s.t. $\epsilon$ is Gaussian-separable.

Denote by $\mathbb{S}$ the feasible solution set for the causal effect of $T$ on $O$, the following Theorem 4 holds.

**Theorem 4.** *Suppose Condition 4 holds. Then, the causal effect of $T$ on $O$ is unidentifiable and $|\mathbb{S}| = \infty$.*

**Intuition.** Under Condition 4, the Gaussian $\epsilon_T$ and the Gaussian components of all Gaussian-separable $\epsilon_{L_i}$ are merged into a unified Gaussian entity. According to Intuition of Corollary 1, this unified Gaussian entity admits infinitely many decompositions into multiple Gaussian components, so $\alpha$ can take infinitely many values.

*Proof.* We prove that given a SCM $\mathcal{M}$ satisfying Condition 4, there exist infinitely many SCMs $\tilde{\mathcal{M}}$ also satisfying Condition 4 s.t. $(T, O) \overset{d}{=} (\tilde{T}, \tilde{O})$ with distinct $\tilde{\alpha}$.

Suppose $\epsilon_{L_1}$ is Gaussian-separable without loss of generality. Based on Corollary 1, there exist $\epsilon'_{L_1}$ and $\epsilon''_{L_1}$ s.t.

$$\epsilon_{L_1} \overset{d}{=} \epsilon'_{L_1} + \epsilon''_{L_1}$$
$$\epsilon'_{L_1} \text{ is Gaussian-separable,}$$
$$\epsilon''_{L_1} \text{ is Gaussian with zero mean,}$$
$$\epsilon'_{L_1} \perp\!\!\!\perp \epsilon''_{L_1} \perp\!\!\!\perp \epsilon_{L_2} \perp\!\!\!\perp \ldots \perp\!\!\!\perp \epsilon_{L_n} \perp\!\!\!\perp \epsilon_T \perp\!\!\!\perp \epsilon_O.$$

Consider a SCM $\tilde{\mathcal{M}}$ in which the number of latent confounders $\tilde{n} = n$:

$$\tilde{L}_1 = \epsilon'_{L_1},$$
$$\tilde{L}_i = \epsilon_{L_i}, \; 2 \leq i \leq n,$$
$$\tilde{T} = \sum_{i=1}^{n} \tilde{L}_i + (\epsilon''_{L_1} + \epsilon_T),$$
$$\tilde{O} = (\alpha + \delta)\tilde{T} + \sum_{i=1}^{n}(\beta_i - \delta)\tilde{L}_i + (\epsilon_O + \epsilon'_O),$$

where

$$\delta = \frac{\text{var}(\epsilon''_{L_1})\beta_1}{\text{var}(\epsilon''_{L_1})+\text{var}(\epsilon_T)},$$
$$\epsilon'_O \sim \mathcal{N}\left(0, \frac{\text{var}(\epsilon''_{L_1})\text{var}(\epsilon_T)\beta_1^2}{\text{var}(\epsilon''_{L_1})+\text{var}(\epsilon_T)}\right),$$
$$\epsilon'_{L_1} \perp\!\!\!\perp \epsilon''_{L_1} \perp\!\!\!\perp \epsilon_{L_2} \perp\!\!\!\perp \ldots \perp\!\!\!\perp \epsilon_{L_n} \perp\!\!\!\perp \epsilon_T \perp\!\!\!\perp \epsilon_O \perp\!\!\!\perp \epsilon'_O.$$

According to Lemma 4 in App. A, there is $(T, O) \overset{d}{=} (\tilde{T}, \tilde{O})$. However, $\tilde{\alpha} = \alpha + \delta \neq \alpha$. Moreover, according to Corollary 1, $\text{var}(\epsilon''_{L_1})$ can have infinitely many possible variances, so $\delta$ can take infinitely many values. $\square$

## § Discussion

We further discuss the testability of Conditions 1–4, that is, whether it is possible to determine which condition holds from the joint distribution of $(T, O)$.

1. If $T$ is Gaussian, then by Cramér's decomposition theorem (Cramér, 1936), $\epsilon_T, \epsilon_{L_1}, \ldots, \epsilon_{L_n}$ are all Gaussian, so Condition 4 holds.

2. If $T$ is not Gaussian-separable, then none of $\epsilon_T, \epsilon_{L_1}, \ldots, \epsilon_{L_n}$ is Gaussian-separable, which implies that all of them are non-Gaussian, so Condition 3 holds.

3. If $T$ is non-Gaussian but Gaussian-separable, then whether $\epsilon_T$ is Gaussian or non-Gaussian is indeterminate, so Conditions 1–4 cannot be distinguished from each other.

*Table 1.* SCMs used to generate synthetic data.

| SCM | $\epsilon_L$ | $\epsilon_T$ | $\epsilon_O$ | Condition |
|---|---|---|---|---|
| $\mathcal{M}_1$ | | Uniform | | |
| $\mathcal{M}_2$ | Gaussian | Exponential | Gaussian | 1 |
| $\mathcal{M}_3$ | | Laplace | | |
| $\mathcal{M}_4$ | Uniform | | | |
| $\mathcal{M}_5$ | Exponential | Gaussian | Gaussian | 2 |
| $\mathcal{M}_6$ | Laplace | | | |

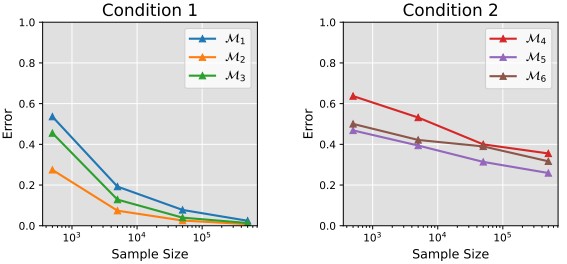

*Figure 3.* Experimental results on $\mathcal{M}_1$–$\mathcal{M}_6$.

In summary, only when $T$ is Gaussian or not Gaussian-separable can we determine which condition holds. Furthermore, it is statistically challenging to reliably test whether $T$ is Gaussian and whether $T$ is Gaussian-separable from a finite sample. Therefore, in practice, it is preferable to rely on problem-specific domain knowledge to determine which of Conditions 1–4 holds.

### 3.2. Empirical Analysis

In this section, we empirically validate our theoretical findings. Specifically, Theorems 1 and 2 guarantee that the causal effect is identifiable, so we generate synthetic data based on Conditions 1 and 2 and estimate the causal effect of $T$ on $O$ from the data.

We consider 6 different SCMs $\mathcal{M}_1$–$\mathcal{M}_6$ with one latent confounder[3], in which the distributions of exogenous noises[4] are summarized in Table 1, where we also indicate which of Conditions 1 and 2 is satisfied in each SCM. For each SCM, we consider sample sizes of 0.5k, 5k, 50k, 500k; for each sample size, we produce 500 datasets. Following Tramontano et al. (2024; 2025); Kivva et al. (2025) with minor modifications, we sample each coefficient from a uniform distribution over $[-2.0, -0.5] \cup [0.5, 2.0]$ and the standard error of each exogenous noise from a uniform distribution over $[0.5, 2.0]$.

We estimate the causal effect of $T$ on $O$ from each dataset.

---

[3] We provide experimental results on SCMs with multiple latent confounders in App. B.2.

[4] Conditions 1 and 2 both impose no restrictions on the distribution of $\epsilon_O$, so we set it to a Gaussian distribution without loss of generality. Experimental results in App. B.1 show that varying the distribution of $\epsilon_O$ does not alter the conclusions.

- In $\mathcal{M}_1$–$\mathcal{M}_3$, assuming $\text{cum}_{4,0}(T, O) \neq 0$[5], we have

$$\alpha = \frac{\text{cum}_{3,1}(T, O)}{\text{cum}_{4,0}(T, O)}$$

based on Eq. (7). Therefore, estimating $\text{cum}_{3,1}(T, O)$ and $\text{cum}_{4,0}(T, O)$ from the given dataset directly yields an estimate of $\alpha$.

- In $\mathcal{M}_4$–$\mathcal{M}_6$, estimating $\alpha$ is essentially equivalent to estimating the noise level in a semiparametric convolution model investigated by Butucea & Matias (2005). Specifically, suppose $X \overset{d}{=} X_1 + X_2$ where $X_1$ is not Gaussian-separable and $X_2$ is Gaussian with zero mean, Butucea & Matias (2005) developed a consistent estimation method for the standard error of $X_2$, which is summarized in Algorithm 1 in App. B. According to Equation (9), let $X = T$, we can obtain an estimate of the standard error of $\epsilon_T$ using Algorithm 1; let $X = O + T$, we can obtain an estimate of the standard error of $(\alpha + 1)\epsilon_T + \epsilon_O''$ using Algorithm 1; and let $X = O - T$, we can obtain an estimate of the standard error of $(\alpha - 1)\epsilon_T + \epsilon_O''$ using Algorithm 1. Note that

$$\alpha = \frac{\text{var}\left((\alpha + 1)\epsilon_T + \epsilon_O''\right) - \text{var}\left((\alpha - 1)\epsilon_T + \epsilon_O''\right)}{4\text{var}(\epsilon_T)},$$

we can readily obtain an estimate of $\alpha$.

Our code is available at: https://github.com/XiuchuanLi/ICML2026-identifiability. The experimental results are summarized in Figure 3. Denote the estimated causal effect by $\hat{\alpha}$, the error is given by $\frac{|\alpha - \hat{\alpha}|}{|\alpha|}$ following Tramontano et al. (2024; 2025); Kivva et al. (2025). Evidently, the estimation errors exhibit a strictly monotonic decrease with increasing sample size, providing empirical support for Theorems 1 and 2. Moreover, with a sample size of 500k, while the estimation errors on $\mathcal{M}_1$–$\mathcal{M}_3$ almost vanish, those for $\mathcal{M}_4$–$\mathcal{M}_6$ remain non-negligible. This phenomenon is consistent with theoretical expectations. Specifically, as rigorously established by Butucea & Matias (2005), the optimal convergence rate of Algorithm 1 is merely logarithmic, which means that achieving high estimation precision requires a prohibitively large sample size. Finally, we include no baseline approach because existing estimation methods typically require auxiliary variables. In particular, although Corollary 2 in Sanjaroonpouri & Ramazi (2025) is related to our Theorem 2 (as discussed in the Remark following it), they do not provide a corresponding estimation method.

---

[5]First, this assumption is mild, as $\text{cum}_{4,0}(T, O) = \text{cum}_4(\epsilon_T)$ and the fourth-order cumulant of a non-Gaussian distribution is typically non-zero unless the distribution is specifically designed to be pathological. Second, this assumption is testable. Specifically, we can set the null hypothesis to $\text{cum}_{4,0}(T, O) = 0$ and evaluate it through a hypothesis test such as a bootstrap-based test. If the null hypothesis cannot be rejected, we simply move to investigate higher-order cumulants

## 4. Related Work

Causal effect identification in the presence of latent confounders has garnered significant attention. Existing studies have established various sufficient conditions that render the causal effect identifiable, most of which entail auxiliary variables subject to specific structural constraints such as mediators (Pearl, 1995), instruments (Angrist et al., 1996), and proxies (Kuroki & Pearl, 2014). While Rothenhäusler et al. (2019); Kivva et al. (2025) have obviated the need for such variables, they necessitate multi-environment distributions. In contrast, we ground identifiability in solely the joint distribution of the treatment-outcome pair (from a single environment).

**§ Mediator.** According to Pearl (1995), with a mediator $Z$ satisfying the three conditions shown in Figure 1(a),

$$P(O = o|do(T = t))$$
$$= \sum_z \sum_{t'} P(O = o|Z = z, T = t')P(T = t')P(Z = z|T = t),$$

from which the causal effect of $T$ on $O$ defined as Equation (3) can be readily derived. Building upon this result, more in-depth investigations have been conducted (Kuroki & Cai, 2004; Glynn & Kashin, 2018; Fulcher et al., 2020; Bhattacharya & Nabi, 2022; Jeong et al., 2022). For instance, Fulcher et al. (2020) generalized this result by relaxing the condition (i) in Figure 1(a). Nevertheless, none of these studies completely obviated the need for mediators, thereby retaining the reliance on auxiliary variables.

**§ Instrument.** According to Angrist et al. (1996), within linear SCMs, given an instrument $Z$ satisfying the three conditions shown in Figure 1(b), the causal effect of $T$ on $O$, i.e. $\alpha$ in Equation (4), is given by

$$\alpha = \frac{\text{cov}(Z, O)}{\text{cov}(Z, T)}.$$

Subsequently, many researchers (Brito & Pearl, 2002; Torgovitsky, 2015; Muandet et al., 2020; Burauel, 2023; Xie et al., 2024b) further advanced this identification paradigm. For instance, Brito & Pearl (2002) proposed a generalized instrument by relaxing condition (iii) in Figure 1(b). Nevertheless, none of these studies completely obviated the need for instruments, thereby retaining the reliance on auxiliary variables. Readers can refer to Wu et al. (2025) for a survey on instrumental causal effect identification.

**§ Proxy.** Kuroki & Pearl (2014) pioneered proximal causal effect identification. They showed that if the four conditions in Figure 1(c) are satisfied, the causal effect of $T$ on $O$ in a linear SCM is identifiable and given by

$$\alpha = \frac{\text{cov}(T, O)\text{cov}(Z_1, Z_2) - \text{cov}(T, Z_2)\text{cov}(O, Z_1)}{\text{var}(T)\text{cov}(Z_1, Z_2) - \text{cov}(T, Z_2)\text{cov}(T, Z_1)}.$$

Also, the causal effect in non-parametric SCMs is identifiable if the four conditions in Figure 1(c) hold and $Z_2 \perp\!\!\!\perp O | L$. This seminal work inspired numerous follow-up studies (Miao et al., 2018; Kivva et al., 2023; Park et al., 2024; Kummerfeld et al., 2024; Xie et al., 2024a). For instance, Kivva et al. (2023) can work with a single proxy. Nevertheless, none of these studies completely obviated the need for proxies, thereby retaining the reliance on auxiliary variables. Readers can refer to Tchetgen Tchetgen et al. (2024) for a review of proximal causal effect identification.

**§ Multi-environment distributions.** To render the causal effect within a linear SCM identifiable, Rothenhäusler et al. (2019); Kivva et al. (2025) leveraged distributions from multiple environments subject to stringent constraints. Specifically, Rothenhäusler et al. (2019) assumed that all coefficients (including but not limited to the causal effect of interest) remain invariant across environments and each exogenous noise in every environment can be decomposed into the sum of an environment-invariant component and an environment-specific component. Kivva et al. (2025) assumed that the causal effect of interest remains invariant across environments, whereas any two environments differ in exactly one coefficient or exogenous noise distribution. Although these two studies obviate the need for auxiliary variables, they trade this for a reliance on multi-environment distributions, whereas we require neither.

## 5. Conclusion

In this paper, we investigate causal effect identifiability in the presence of latent confounders without auxiliary variables. Specifically, we ground identifiability in solely the joint distribution of the treatment-effect pair, which constitutes the irreducible statistical basis of causal effect identification. Focusing on linear SCMs, we formulate a set of mutually exclusive and collectively exhaustive conditions regarding the Gaussianity of exogenous noises, ascertain the identifiability status under each of them, and also quantifies the cardinality of the feasible solution set. Our results fill a significant gap in the identifiability theory. Finally, we empirically validate our theoretical findings.

While our contributions are rigorously established in linear SCMs, the core insights may extend to nonlinear settings. We provide some preliminary theoretical results that establish identifiability of the causal effect in certain nonlinear scenarios in the App. C, leaving a more comprehensive investigation for future research.

## Acknowledgments

Tongliang Liu is partially supported by the following Australian Research Council projects: FT220100318, DP 260102466, DP220102121, LP220100527, LP220200949. James Kwok is supported in part by the Research Grants Council of the Hong Kong Special Administrative Region (Grants 16202523 and HKU C7004-22G). Xiu-Chuan Li is partially supported by ARC FT220100318 and Google PhD Fellowship. Xiu-Chuan would like to thank Suqin Yuan for his helpful advice.

## Impact Statement

This paper presents work whose goal is to advance the field of causal inference. There are many potential societal consequences of our work, none of which we feel must be specifically highlighted here.

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

**Table of Contents**

## A. Proofs of Lemmas and Corollaries

**Lemma 1.**  If $X$ is Gaussian-separable, then there exist $X_1$ and $X_2$ s.t. (i) $X \overset{d}{=} X_1 + X_2$, (ii) $X_1 \perp\!\!\!\perp X_2$, (iii) $X_1$ is not Gaussian-separable, and (iv) $X_2$ is Gaussian with zero mean. Moreover, the distributions of $X_1$ and $X_2$ are unique.

*Proof.* Let $S = \{\sigma'^2 | X \overset{d}{=} X_1' + X_2', X_1' \perp\!\!\!\perp X_2', X_2' \sim \mathcal{N}(0, \sigma'^2)\}$ and $\sigma_{\max}^2 = \sup S$. Based on the Lévy's continuity theorem (Williams, 1991), $\sigma_{\max}^2 \in S$. Let

$$X \overset{d}{=} X_1 + X_2,$$
$$X_1 \perp\!\!\!\perp X_2,$$
$$X_2 \sim \mathcal{N}(0, \sigma_{\max}^2),$$

we can prove that $X_1$ is not Gaussian-separable by contradiction. Specifically, suppose $X_1$ is Gaussian-separable, let

$$X_1 \overset{d}{=} Y_1 + Y_2,$$
$$Y_1 \perp\!\!\!\perp Y_2 \perp\!\!\!\perp X_2,$$
$$Y_2 \text{ is Gaussian with zero mean.}$$

It follows that $X \overset{d}{=} Y_1 + Y_2 + X_2$ where $Y_2 + X_2$ is Gaussian with zero mean but $\mathrm{var}(Y_2 + X_2) > \sigma_{\max}^2$, which leads to a contradiction.

Moreover, since $X_2 \sim \mathcal{N}(0, \sigma_{\max}^2)$, the distribution of $X_2$ is unique. In addition, since $X \overset{d}{=} X_1 + X_2$ and $X_1 \perp\!\!\!\perp X_2$, $\phi_{X_1}(t)\phi_{X_2}(t) = \phi_X(t)$, that is, the distribution of $X_1$ is also unique. $\qquad\square$

**Corollary 1.**  If $X$ is Gaussian-separable, then there exist $X_1', X_2'$ s.t. (i) $X \overset{d}{=} X_1' + X_2'$, (ii) $X_1' \perp\!\!\!\perp X_2'$, (iii) $X_1'$ is Gaussian-separable, and (iv) $X_2$ is Gaussian with zero mean. Moreover, the distributions of $X_1'$ and $X_2'$ can take infinitely many forms.

*Proof.* Suppose $X_1$ and $X_2$ satisfy the conditions in Lemma 1. Then, given $\sigma'^2 \in (0, \mathrm{var}(X_2))$, let

$$Y_1 \sim \mathcal{N}(0, \mathrm{var}(X_2) - \sigma'^2),$$
$$Y_2 \sim \mathcal{N}(0, \sigma'^2),$$
$$X_1 \perp\!\!\!\perp Y_1 \perp\!\!\!\perp Y_2.$$

It follows that $X_1 + Y_1 + Y_2 \overset{d}{=} X$. Let $X_1' = X_1 + Y_1$ and $X_2' = Y_2$, it is clear that conditions (i)-(iv) are satisfied.

Moreover, the variance of $X_2'$ can take any value within $(0, \mathrm{var}(X_2))$, implying that the distribution of $X_2'$ can take infinitely many forms. Correspondingly, the distribution of $X_1'$ can also take infinitely many forms. $\qquad\square$

**Lemma 2.**  For any integers $i > 0$ and $j \geq 0$,

$$\mathrm{cum}_{i,j}(T, O) = \sum_{k=1}^{n} (\alpha + \beta_k)^j \mathrm{cum}_{i+j}(\epsilon_{L_k}) + \alpha^j \mathrm{cum}_{i+j}(\epsilon_T).$$

*Proof.* Wring $T, O$ as linear combinations of exogenous noises that are mutually independent:

$$T = \sum_{k=1}^{n} \epsilon_{L_k} + \epsilon_T,$$

$$O = \sum_{k=1}^{n} (\alpha + \beta_k) \epsilon_{L_k} + \alpha \epsilon_T + \epsilon_O,$$

we have

$$\text{cum}_{i,j}(T, O) = \text{cum}_{i,j} \left( \sum_{k=1}^{n} \epsilon_{L_k} + \epsilon_T, \sum_{k=1}^{n} (\alpha + \beta_k) \epsilon_{L_k} + \alpha \epsilon_T + \epsilon_O \right)$$

$$= \sum_{k=1}^{n} \text{cum}_{i,j} \left( \epsilon_{L_k}, (\alpha + \beta_k) \epsilon_{L_k} \right) + \text{cum}_{i,j}(\epsilon_T, \alpha \epsilon_T) + \text{cum}_{i,j}(0, \epsilon_O)$$

$$= \sum_{k=1}^{n} (\alpha + \beta_k)^j \text{cum}_{i+j}(\epsilon_{L_k}) + \alpha^j \text{cum}_{i+j}(\epsilon_T) + 0. \tag{11}$$

This completes the proof. □

**Remark.** Notably, the validity of Lemma 2 does not rely on any of Conditions 1–4.

$A(X, Y; p, q, r) \in \mathbb{R}^{p \times p}$ is composed of cumulants of $X$ and $Y$ where $p, q, r$ can be arbitrary integers subject to

$$p \geq 1, \ q \geq 0, \ r \geq 2p + q - 1.$$

The $(i, j)$-th element of $A(X, Y; p, q, r)$ is given by

$$A(X, Y; p, q, r)_{i,j} = \text{cum}_{r-(q+i+j-2), q+i+j-2}(X, Y).$$

Specifically,

$$A(X, Y; p, q, r)_{i,j} = \begin{bmatrix} \text{cum}_{r-q,q}(X,Y) & \text{cum}_{r-(q+1),q+1}(X,Y) & \cdots & \text{cum}_{r-(q+p-1),q+p-1}(X,Y) \\ \text{cum}_{r-(q+1),q+1}(X,Y) & \text{cum}_{r-(q+2),q+2}(X,Y) & \cdots & \text{cum}_{r-(q+p),q+p}(X,Y) \\ \vdots & \vdots & \ddots & \vdots \\ \text{cum}_{r-(q+p-1),q+p-1}(X,Y) & \text{cum}_{r-(q+p),q+p}(X,Y) & \cdots & \text{cum}_{r-(q+2p-2),q+2p-2}(X,Y) \end{bmatrix}.$$

For instance,

$$A(X, Y; 3, 1, 7) = \begin{bmatrix} \text{cum}_{6,1}(X,Y) & \text{cum}_{5,2}(X,Y) & \text{cum}_{4,3}(X,Y) \\ \text{cum}_{5,2}(X,Y) & \text{cum}_{4,3}(X,Y) & \text{cum}_{3,4}(X,Y) \\ \text{cum}_{4,3}(X,Y) & \text{cum}_{3,4}(X,Y) & \text{cum}_{2,5}(X,Y) \end{bmatrix}.$$

**Lemma 3.** Let $m$ denote $|\{\epsilon_{L_i}|\epsilon_{L_i} \text{ is non-Gaussian}\}|$ and $m'$ denote $\mathbb{1}(\epsilon_T \text{ is non-Gaussian})$ where $\mathbb{1}(\cdot)$ is the indicator function. Then, for any $p \geq \max(m + m', 1)$ and $r \geq \max(2p, 3)$, the rank of $A(T, O; p, 1, r) - \lambda A(T, O; p, 0, r)$ is less than the rank of $A(T, O; p, 0, r)$ if and only if

$$\lambda \in \begin{cases} \{\alpha + \beta_i | \text{cum}_r(\epsilon_{L_i}) \neq 0\} & \text{if } \text{cum}_r(\epsilon_T) = 0 \\ \{\alpha + \beta_i | \text{cum}_r(\epsilon_{L_i}) \neq 0\} \cup \{\alpha\} & \text{if } \text{cum}_r(\epsilon_T) \neq 0 \end{cases}.$$

*Proof.* If $\{\epsilon_{L_i}|\epsilon_{L_i} \text{ is non-Gaussian}\} = \emptyset$, then $A(T, O; p, 1, r) = \alpha A(T, O; p, 0, r)$ and $\text{rk}(A(T, O; p, 1, r)) = 1$ if $\text{cum}_r(\epsilon_T) \neq 0$; $A(T, O; p, 1, r) = A(T, O; p, 0, r) = \mathbf{0}_{p \times p}$ if $\text{cum}_r(\epsilon_T) = 0$, so it is trivial that Lemma 3 holds.

If $\{\epsilon_{L_i}|\epsilon_{L_i} \text{ is non-Gaussian}\} \neq \emptyset$, without loss of generality, we assume $\{\epsilon_{L_i}|\epsilon_{L_i} \text{ is non-Gaussian}\} = \{\epsilon_{L_1}, ..., \epsilon_{L_m}\}$. Let

$$c_i^{(r)} = \text{cum}_r(\epsilon_{L_i}), \quad \zeta_i = \alpha + \beta_i, \quad 1 \leq i \leq m.$$

If $m' = 1$, let

$$c_{m+1}^{(r)} = \mathrm{cum}_r(\epsilon_T), \quad \zeta_{m+1} = \alpha.$$

According to Lemma 2, for any integers $u > 0$ and $v \geq 0$ s.t. $u + v > 2$, we have

$$\mathrm{cum}_{u,v}(T, O) = \sum_{i=1}^{m+m'} \zeta_i^v c_i^{(u+v)}.$$

Because $r > 2$,

$$
A(T, O; p, 1, r) = 
\begin{bmatrix}
\sum_{i=1}^{m+m'} \zeta_i c_i^{(r)} & \sum_{i=1}^{m+m'} \zeta_i^2 c_i^{(r)} & \cdots & \sum_{i=1}^{m+m'} \zeta_i^p c_i^{(r)} \\
\sum_{i=1}^{m+m'} \zeta_i^2 c_i^{(r)} & \sum_{i=1}^{m+m'} \zeta_i^3 c_i^{(r)} & \cdots & \sum_{i=1}^{m+m'} \zeta_i^{p+1} c_i^{(r)} \\
\vdots & \vdots & \ddots & \vdots \\
\sum_{i=1}^{m+m'} \zeta_i^p c_i^{(r)} & \sum_{i=1}^{m+m'} \zeta_i^{p+1} c_i^{(r)} & \cdots & \sum_{i=1}^{m+m'} \zeta_i^{2p-1} c_i^{(r)}
\end{bmatrix}
$$

$$
= 
\begin{bmatrix}
1 & 1 & \cdots & 1 \\
\zeta_1 & \zeta_2 & \cdots & \zeta_{m+m'} \\
\vdots & \vdots & \ddots & \vdots \\
\zeta_1^{p-1} & \zeta_2^{p-1} & \cdots & \zeta_{m+m'}^{p-1}
\end{bmatrix}
\mathrm{diag}\left(
\begin{bmatrix}
\zeta_1 c_1^{(r)} \\
\zeta_2 c_2^{(r)} \\
\vdots \\
\zeta_{m+m'} c_{m+m'}^{(r)}
\end{bmatrix}
\right)
\begin{bmatrix}
1 & \zeta_1 & \cdots & \zeta_1^{p-1} \\
1 & \zeta_2 & \cdots & \zeta_2^{p-1} \\
\vdots & \vdots & \ddots & \vdots \\
1 & \zeta_{m+m'} & \cdots & \zeta_{m+m'}^{p-1}
\end{bmatrix}
$$

and

$$
A(T, O; p, 0, r) = 
\begin{bmatrix}
\sum_{i=1}^{m+m'} c_i^{(r)} & \sum_{i=1}^{m+m'} \zeta_i c_i^{(r)} & \cdots & \sum_{i=1}^{m+m'} \zeta_i^{p-1} c_i^{(r)} \\
\sum_{i=1}^{m+m'} \zeta_i c_i^{(r)} & \sum_{i=1}^{m+m'} \zeta_i^2 c_i^{(r)} & \cdots & \sum_{i=1}^{m+m'} \zeta_i^p c_i^{(r)} \\
\vdots & \vdots & \ddots & \vdots \\
\sum_{i=1}^{m+m'} \zeta_i^{p-1} c_i^{(r)} & \sum_{i=1}^{m+m'} \zeta_i^p c_i^{(r)} & \cdots & \sum_{i=1}^{m+m'} \zeta_i^{2p-2} c_i^{(r)}
\end{bmatrix}
$$

$$
= 
\begin{bmatrix}
1 & 1 & \cdots & 1 \\
\zeta_1 & \zeta_2 & \cdots & \zeta_{m+m'} \\
\vdots & \vdots & \ddots & \vdots \\
\zeta_1^{p-1} & \zeta_2^{p-1} & \cdots & \zeta_{m+m'}^{p-1}
\end{bmatrix}
\mathrm{diag}\left(
\begin{bmatrix}
c_1^{(r)} \\
c_2^{(r)} \\
\vdots \\
c_{m+m'}^{(r)}
\end{bmatrix}
\right)
\begin{bmatrix}
1 & \zeta_1 & \cdots & \zeta_1^{p-1} \\
1 & \zeta_2 & \cdots & \zeta_2^{p-1} \\
\vdots & \vdots & \ddots & \vdots \\
1 & \zeta_{m+m'} & \cdots & \zeta_{m+m'}^{p-1}
\end{bmatrix},
$$

so

$$
A(T, O; p, 1, r) - \lambda A(T, O; p, 0, r) = 
\begin{bmatrix}
1 & 1 & \cdots & 1 \\
\zeta_1 & \zeta_2 & \cdots & \zeta_{m+m'} \\
\vdots & \vdots & \ddots & \vdots \\
\zeta_1^{p-1} & \zeta_2^{p-1} & \cdots & \zeta_{m+m'}^{p-1}
\end{bmatrix}
\mathrm{diag}\left(
\begin{bmatrix}
(\zeta_1 - \lambda) c_1^{(r)} \\
(\zeta_2 - \lambda) c_2^{(r)} \\
\vdots \\
(\zeta_{m+m'} - \lambda) c_{m+m'}^{(r)}
\end{bmatrix}
\right)
\begin{bmatrix}
1 & \zeta_1 & \cdots & \zeta_1^{p-1} \\
1 & \zeta_2 & \cdots & \zeta_2^{p-1} \\
\vdots & \vdots & \ddots & \vdots \\
1 & \zeta_{m+m'} & \cdots & \zeta_{m+m'}^{p-1}
\end{bmatrix}.
$$

Note that the $(m + m') \times (m + m')$ leading principal submatrix of $\begin{bmatrix} 1 & \zeta_1 & \cdots & \zeta_1^{p-1} \\ 1 & \zeta_2 & \cdots & \zeta_2^{p-1} \\ \vdots & \vdots & \ddots & \vdots \\ 1 & \zeta_{m+m'} & \cdots & \zeta_{m+m'}^{p-1} \end{bmatrix}$ is a Vandermonde matrix and

$\zeta_i \neq \zeta_j$ for any $i \neq j$ based on Assumption 1, so its $(m + m') \times (m + m')$ leading principal submatrix is invertible, which

means that it has full row rank. Therefore,

$$\text{rk}(A(T,O;p,0,r)) = \text{rk}\left(\text{diag}\left(\begin{bmatrix} c_1^{(r)} \\ c_2^{(r)} \\ \vdots \\ c_{m+m'}^{(r)} \end{bmatrix}\right)\right),$$

$$\text{rk}(A(T,O;p,1,r) - \lambda A(T,O;p,0,r)) = \text{rk}\left(\text{diag}\left(\begin{bmatrix} (\zeta_1 - \lambda)c_1^{(r)} \\ (\zeta_2 - \lambda)c_2^{(r)} \\ \vdots \\ (\zeta_{m+m'} - \lambda)c_{m+m'}^{(r)} \end{bmatrix}\right)\right),$$

this completes the proof. $\qquad\square$

**Remark.** Notably, the validity of Lemma 3 does not rely on any of Conditions 1–4.

**Condition 3.** $\epsilon_T$ is non-Gaussian and there exists $\epsilon \in \{\epsilon_{L_1}, ..., \epsilon_{L_n}\}$ s.t. $\epsilon$ is non-Gaussian.

**Corollary 2.** *Suppose Condition 3 holds and let $m$ denote $|\{\epsilon_{L_i}|\epsilon_{L_i} \text{ is non-Gaussian}\}|$. Then, $m$ can be uniquely determined by the joint distribution of $(T,O)$.*

*Proof.* We prove this lemma by contradiction. Suppose there exist two SCMs $\mathcal{M}$ and $\tilde{\mathcal{M}}$ satisfying Condition 3 s.t. $(T,O) \overset{d}{=} (\tilde{T},\tilde{O})$ and $m \neq \tilde{m}$. Without loss of generality, we assume $m > \tilde{m}$ and $\{\epsilon_{L_i}|\epsilon_{L_i} \text{ is non-Gaussian}\} = \{\epsilon_{L_1}, ..., \epsilon_{L_m}\}$. For any $\epsilon_{L_i} \in \{\epsilon_{L_1}, ..., \epsilon_{L_m}\}$, there exists $r_i \geq 2m + 2$ s.t. $\text{cum}(\epsilon_{L_i}) \neq 0$. Furthermore, there also exists $r_{m+1} \geq 2m + 2$ s.t. $\text{cum}(\epsilon_T) \neq 0$. Let

$$\Lambda = \bigcup_{i=1}^{m+1} \{\lambda|\text{rk}(A(T,O;m+1,1,r_i) - \lambda A(T,O;m+1,0,r_i)) < \text{rk}(A(T,O;m+1,0,r_i))\},$$

$$\tilde{\Lambda} = \bigcup_{i=1}^{m+1} \{\lambda|\text{rk}(A(\tilde{T},\tilde{O};m+1,1,r_i) - \lambda A(\tilde{T},\tilde{O};m+1,0,r_i)) < \text{rk}(A(\tilde{T},\tilde{O};m+1,0,r_i))\}.$$

Based on Lemma 3, we have $|\Lambda| = m + 1$ and $|\tilde{\Lambda}| \leq \tilde{m} + 1$. However, because $(T,O) \overset{d}{=} (\tilde{T},\tilde{O})$, we also have $\tilde{\Lambda} = \Lambda$, which leads to a contradiction. $\qquad\square$

**Condition 4.** $\epsilon_T$ is Gaussian and there exists $\epsilon \in \{\epsilon_{L_1}, ..., \epsilon_{L_n}\}$ s.t. $\epsilon$ is Gaussian-separable.

**Lemma 4.** *Given a SCM $\mathcal{M}$:*

$$L_i = \epsilon_{L_i}, \ 1 \leq i \leq n,$$
$$T = \sum_{i=1}^{n} L_i + \epsilon_T,$$
$$O = \alpha T + \sum_{i=1}^{n} \beta_i L_i + \epsilon_O,$$

*that satisfies Condition 4, and another SCM $\tilde{\mathcal{M}}$ in which $\tilde{n} = n$:*

$$\tilde{L}_1 = \epsilon'_{L_1},$$
$$\tilde{L}_i = \epsilon_{L_i}, \ 2 \leq i \leq n,$$
$$\tilde{T} = \sum_{i=1}^{n} \tilde{L}_i + (\epsilon''_{L_1} + \epsilon_T),$$
$$\tilde{O} = (\alpha + \delta)\tilde{T} + \sum_{i=1}^{n} (\beta_i - \delta)\tilde{L}_i + (\epsilon_O + \epsilon'_O),$$

*where*

$$\epsilon_{L_1} \overset{d}{=} \epsilon'_{L_1} + \epsilon''_{L_1}$$
$$\epsilon'_{L_1} \text{ is Gaussian-separable,}$$
$$\epsilon''_{L_1} \text{ is Gaussian with zero mean,}$$
$$\delta = \frac{\text{var}(\epsilon''_{L_1})\beta_1}{\text{var}(\epsilon''_{L_1}) + \text{var}(\epsilon_T)},$$
$$\epsilon'_O \sim \mathcal{N}\left(0, \frac{\text{var}(\epsilon''_{L_1})\text{var}(\epsilon_T)\beta_1^2}{\text{var}(\epsilon''_{L_1}) + \text{var}(\epsilon_T)}\right),$$
$$\epsilon'_{L_1} \perp\!\!\!\perp \epsilon''_{L_1} \perp\!\!\!\perp \epsilon_{L_2} \perp\!\!\!\perp ... \perp\!\!\!\perp \epsilon_{L_n} \perp\!\!\!\perp \epsilon_T \perp\!\!\!\perp \epsilon_O \perp\!\!\!\perp \epsilon'_O.$$

*there is*

$$\phi_{(T,O)}((t_1, t_2)) = \phi_{(\tilde{T},\tilde{O})}((t_1, t_2)).$$

*Proof.* Wring $T, O$ as linear combinations of exogenous noises that are mutually independent:

$$T = \sum_{i=1}^{n} \epsilon_{L_i} + \epsilon_T,$$

$$O = \sum_{i=1}^{n} (\alpha + \beta_i)\epsilon_{L_i} + \alpha\epsilon_T + \epsilon_O,$$

we have

$$\begin{aligned}
&\phi_{(T,O)}((t_1, t_2))\\
=&\mathbb{E}[\exp(jt_1 T + jt_2 O)]\\
=&\mathbb{E}[\exp(\sum_{i=1}^{n} j(t_1 + (\alpha + \beta_i)t_2)\epsilon_{L_i} + j(t_1 + \alpha t_2)\epsilon_T + jt_2\epsilon_O)]\\
=&\phi_{\epsilon_T}(t_1 + \alpha t_2)\phi_{\epsilon_O}(t_2)\prod_{i=1}^{n}\phi_{\epsilon_{L_i}}(t_1 + (\alpha + \beta_i)t_2)\\
=&\phi_{\epsilon_T}(t_1 + \alpha t_2)\phi_{\epsilon_O}(t_2)\phi_{\epsilon'_{L_1}}(t_1 + (\alpha + \beta_1)t_2)\phi_{\epsilon''_{L_1}}(t_1 + (\alpha + \beta_1)t_2)\prod_{i=2}^{n}\phi_{\epsilon_{L_i}}(t_1 + (\alpha + \beta_i)t_2).
\end{aligned}$$

Wring $\tilde{T}, \tilde{O}$ as linear combinations of exogenous noises that are mutually independent:

$$\tilde{T} = \epsilon'_{L_1} + \epsilon''_{L_1} + \sum_{i=2}^{n} \epsilon_{L_i} + \epsilon_T,$$

$$\tilde{O} = (\alpha + \beta_1)\epsilon'_{L_1} + (\alpha + \delta)\epsilon''_{L_1} + \sum_{i=2}^{n} (\alpha + \beta_i)\epsilon_{L_i} + (\alpha + \delta)\epsilon_T + \epsilon'_O + \epsilon_O,$$

we have

$$\begin{aligned}
&\phi_{(\tilde{T},\tilde{O})}((t_1, t_2))\\
=&\mathbb{E}[\exp(jt_1\tilde{T} + jt_2\tilde{O})]\\
=&\mathbb{E}[\exp(j(t_1 + (\alpha + \beta_1)t_2)\epsilon'_{L_1} + j(t_1 + (\alpha + \delta)t_2)\epsilon''_{L_1} + \sum_{i=2}^{n} j(t_1 + (\alpha + \beta_i)t_2)\epsilon_{L_i} + j(t_1 + (\alpha + \delta)t_2)\epsilon_T + jt_2\epsilon'_O + jt_2\epsilon_O)]\\
=&\phi_{\epsilon_T}(t_1 + (\alpha + \delta)t_2)\phi_{\epsilon'_O}(t_2)\phi_{\epsilon_O}(t_2)\phi_{\epsilon'_{L_1}}(t_1 + (\alpha + \beta_1)t_2)\phi_{\epsilon''_{L_1}}(t_1 + (\alpha + \delta)t_2)\prod_{i=2}^{n}\phi_{\epsilon_{L_i}}(t_1 + (\alpha + \beta_i)t_2).
\end{aligned}$$

Clearly, to prove $\phi_{(T,O)}((t_1, t_2)) = \phi_{(\tilde{T},\tilde{O})}((t_1, t_2))$, we only need to prove

$$\phi_{\epsilon''_{L_1}}(t_1 + (\alpha + \beta_1)t_2)\phi_{\epsilon_T}(t_1 + \alpha t_2) = \phi_{\epsilon''_{L_1}}(t_1 + (\alpha + \delta)t_2)\phi_{\epsilon_T}(t_1 + (\alpha + \delta)t_2)\phi_{\epsilon'_O}(t_2),$$

which is equivalent to

$$\phi_{(\epsilon''_{L_1} + \epsilon_T, (\alpha+\beta_1)\epsilon''_{L_1} + \alpha\epsilon_T)}((t_1, t_2)) = \phi_{(\epsilon''_{L_1} + \epsilon_T, (\alpha+\delta)\epsilon''_{L_1} + (\alpha+\delta)\epsilon_T + \epsilon'_O)}((t_1, t_2)). \tag{12}$$

Note that the joint distribution of $(\epsilon''_{L_1} + \epsilon_T, (\alpha + \beta_1)\epsilon''_{L_1} + \alpha\epsilon_T)$ and that of $(\epsilon''_{L_1} + \epsilon_T, (\alpha + \delta)\epsilon''_{L_1} + (\alpha + \delta)\epsilon_T + \epsilon'_O)$ are both

Gaussian, because

$$
\begin{aligned}
&\mathrm{cov}(\epsilon''_{L_1} + \epsilon_T, (\alpha + \delta)\epsilon''_{L_1} + (\alpha + \delta)\epsilon_T + \epsilon'_O) \\
&= (\alpha + \delta)\mathrm{var}(\epsilon''_{L_1}) + (\alpha + \delta)\mathrm{var}(\epsilon_T) \\
&= \left(\alpha + \frac{\mathrm{var}(\epsilon''_{L_1})\beta_1}{\mathrm{var}(\epsilon''_{L_1}) + \mathrm{var}(\epsilon_T)}\right)\left(\mathrm{var}(\epsilon''_{L_1}) + \mathrm{var}(\epsilon_T)\right) \\
&= (\alpha + \beta_1)\mathrm{var}(\epsilon''_{L_1}) + \alpha\mathrm{var}(\epsilon_T) \\
&= \mathrm{cov}(\epsilon''_{L_1} + \epsilon_T, (\alpha + \beta_1)\epsilon''_{L_1} + \alpha\epsilon_T)
\end{aligned}
$$

and

$$
\begin{aligned}
&\mathrm{var}((\alpha + \delta)\epsilon''_{L_1} + (\alpha + \delta)\epsilon_T + \epsilon'_O) \\
&= (\alpha^2 + 2\alpha\delta + \delta^2)\mathrm{var}(\epsilon''_{L_1}) + (\alpha^2 + 2\alpha\delta + \delta^2)\mathrm{var}(\epsilon_T) + \mathrm{var}(\epsilon'_O) \\
&= \left(\alpha^2 + 2\frac{\mathrm{var}(\epsilon''_{L_1})\beta_1}{\mathrm{var}(\epsilon''_{L_1}) + \mathrm{var}(\epsilon_T)} + \frac{\left(\mathrm{var}(\epsilon''_{L_1})\beta_1\right)^2}{\left(\mathrm{var}(\epsilon''_{L_1}) + \mathrm{var}(\epsilon_T)\right)^2}\right)\left(\mathrm{var}(\epsilon''_{L_1}) + \mathrm{var}(\epsilon_T)\right) + \frac{\mathrm{var}(\epsilon''_{L_1})\mathrm{var}(\epsilon_T)\beta_1^2}{\mathrm{var}(\epsilon''_{L_1}) + \mathrm{var}(\epsilon_T)} \\
&= (\alpha^2 + 2\alpha\beta_1 + \beta_1^2)\mathrm{var}(\epsilon''_{L_1}) + \alpha^2\mathrm{var}(\epsilon_T) \\
&= \mathrm{var}((\alpha + \beta_1)\epsilon''_{L_1} + \alpha\epsilon_T),
\end{aligned}
$$

Equation (12) holds. This completes the proof. □

# B. Experiments

Suppose $X \stackrel{d}{=} X_1 + X_2$ where $X_1$ is not Gaussian-separable and $X_2$ is Gaussian with zero mean, Butucea & Matias (2005) developed a consistent estimation method for the standard error of $X_2$, which is summarized as Algorithm 1 below.

---

**Algorithm 1** Estimating the noise level in a semiparametric convolution model (Butucea & Matias, 2005)

---

1: **Input:** $n$ i.i.d. observations of $X$: $x_1, ..., x_n$
2: **Output:** An estimate of the standard error of $X_2$.
3: $\hat{\sigma} := \sqrt{\frac{\sum_{i=1}^n x_i^2}{2n}}$.
4: **while True do**
5:     $u_n := \sqrt{\frac{\log n}{2\hat{\sigma}^2}}$
6:     $\hat{\phi}_n(u_n) := \frac{1}{n}\sum_{i=1}^n \exp(ju_n x_i)$
7:     $\hat{\sigma}' := \frac{\sqrt{-\log|\hat{\phi}_n(u_n)|}}{u_n}$
8:     **if** $|\hat{\sigma} - \hat{\sigma}'| < 1 \times 10^{-4}$ **then**
9:         **break**
10:     **end if**
11:     $\hat{\sigma} := \hat{\sigma}'$.
12: **end while**
13: **return** $\sqrt{2}\hat{\sigma}$

---

## B.1. Varying Distributions of $\epsilon_O$

We consider 6 different SCMs $\mathcal{M}_1^a$–$\mathcal{M}_6^a$ where the distributions of exogenous noises are summarized in Table 2. Clearly, $\mathcal{M}_i^a$ differs from $\mathcal{M}_i$ only in $\epsilon_O$. The experimental results on $\mathcal{M}_1^a$–$\mathcal{M}_6^a$ are summarized in Figure 4, which are consistent with those on $\mathcal{M}_1$–$\mathcal{M}_6$.

*Table 2.* SCMs with varying distributions of $\epsilon_O$.

| SCM | $\epsilon_L$ | $\epsilon_T$ | $\epsilon_O$ | Condition |
|---|---|---|---|---|
| $\mathcal{M}_1^a$ | | Uniform | Uniform | |
| $\mathcal{M}_2^a$ | Gaussian | Exponential | Exponential | 1 |
| $\mathcal{M}_3^a$ | | Laplace | Laplace | |
| $\mathcal{M}_4^a$ | Uniform | | Uniform | |
| $\mathcal{M}_5^a$ | Exponential | Gaussian | Exponential | 2 |
| $\mathcal{M}_6^a$ | Laplace | | Laplace | |

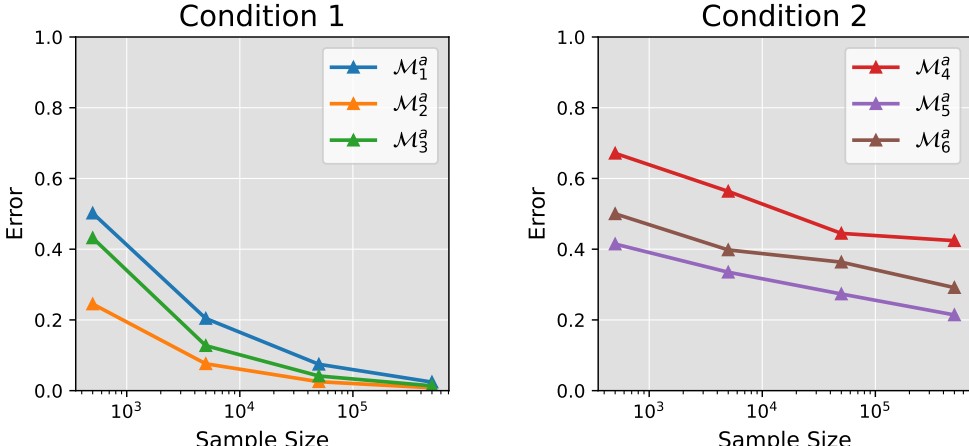

*Figure 4.* Experimental results on $\mathcal{M}_1^a$–$\mathcal{M}_6^a$ with varying distributions of $\epsilon_O$.

### B.2. Multiple Latent Confounders

We consider 6 different SCMs $\mathcal{M}_1^b$–$\mathcal{M}_6^b$ with two latent confounders where the distributions of exogenous noises are summarized in Table 3. The experimental results on $\mathcal{M}_1^b$–$\mathcal{M}_6^b$ are summarized in Figure 5, which are consistent with those on $\mathcal{M}_1$–$\mathcal{M}_6$.

*Table 3.* SCMs with multiple latent confounders.

| SCM | $\epsilon_{L_1}$ | $\epsilon_{L_2}$ | $\epsilon_T$ | $\epsilon_O$ | Condition |
|---|---|---|---|---|---|
| $\mathcal{M}_1^b$ | | | Uniform | | |
| $\mathcal{M}_2^b$ | | Gaussian | Exponential | Gaussian | 1 |
| $\mathcal{M}_3^b$ | | | Laplace | | |
| $\mathcal{M}_4^b$ | | Uniform | | | |
| $\mathcal{M}_5^b$ | | Exponential | Gaussian | Gaussian | 2 |
| $\mathcal{M}_6^b$ | | Laplace | | | |

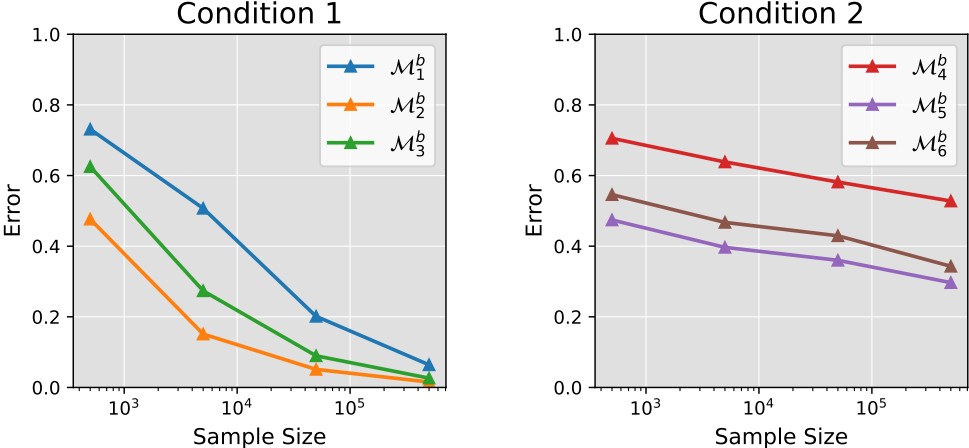

*Figure 5.* Experimental results on $\mathcal{M}_1^b$–$\mathcal{M}_6^b$ with multiple latent confounders.

## C. Extension to Nonlinear Scenarios

In the main text, focusing on linear SCMs, we provide a nuanced and complete characterization of identifiability of the causal effect. In this section, inspired by Theorems 1 and 2, we present some preliminary results that establish identifiability of the causal effect in certain nonlinear scenarios. Specifically, following Guo et al. (2025), we focus on the Additive NonlInear, Constant Effects (ANICE) models where

$$
\begin{aligned}
L_i &= \epsilon_{L_i}, \ 1 \leq i \leq n, \\
T &= f_T(L_1, ..., L_n) + \epsilon_T, \\
O &= \alpha T + f_O(L_1, ..., L_n) + \epsilon_O.
\end{aligned}
\tag{13}
$$

Within such a model, the causal effect of $T$ on $O$ defined as Equation (3) equals $\alpha$. Based on Equation (13), $L_i, T, O$ can also be written as

$$
\begin{aligned}
L_i &= \epsilon_{L_i}, \ 1 \leq i \leq n, \\
T &= f_T(\epsilon_{L_1}, ..., \epsilon_{L_n}) + \epsilon_T, \\
O &= \alpha f_T(\epsilon_{L_1}, ..., \epsilon_{L_n}) + f_O(\epsilon_{L_1}, ..., \epsilon_{L_n}) + \alpha \epsilon_T + \epsilon_O.
\end{aligned}
\tag{14}
$$

Inspired by Theorem 1, we have

**Condition 5.** $\epsilon_T$ is non-Gaussian and any non-zero linear combination of $f_T(\epsilon_{L_1}, ..., \epsilon_{L_n})$ and $f_O(\epsilon_{L_1}, ..., \epsilon_{L_n})$ is Gaussian.

**Corollary 3.** *Suppose Condition 5 holds. Then, the causal effect of $T$ on $O$ is identifiable.*

*Proof.* Similar to Equation (11) in the proof of Lemma 2, for any integers $i > 0$ and $j \geq 0$, we have

$$
\mathrm{cum}_{i,j}(T, O) = \mathrm{cum}_{i,j}\left(f_T(\epsilon_{L_1}, ..., \epsilon_{L_n}), \alpha f_T(\epsilon_{L_1}, ..., \epsilon_{L_n}) + f_O(\epsilon_{L_1}, ..., \epsilon_{L_n})\right) + \alpha^j \mathrm{cum}_{i+j}(\epsilon_T).
$$

Because of Condition 5, if $i + j > 2$, we have

$$
\mathrm{cum}_{i,j}(T, O) = \alpha^j \mathrm{cum}_{i+j}(\epsilon_T).
\tag{15}
$$

The subsequent proof proceeds analogously to the proof of Theorem 1. □

**Remark.** A typical scenario where Condition 5 holds is when $f_T$ and $f_O$ are both linear functions and Condition 1 is satisfied. In addition, Condition 5 also holds when $\epsilon_T$ is non-Gaussian, $\epsilon_{L_1}$ and $\epsilon_{L_2}$ follow uniform distributions, and $f_T$ and $f_O$ take the specific nonlinear forms described by Box & Muller (1958).

Inspired by Theorem 2, we have

**Condition 6.** $\epsilon_T$ is Gaussian and any linear combination of $f_T(\epsilon_{L_1}, ..., \epsilon_{L_n})$ and $f_O(\epsilon_{L_1}, ..., \epsilon_{L_n})$ is not Gaussian-separable.

**Corollary 4.** *Suppose Condition 6 holds. Then, the causal effect of T on O is identifiable.*

*Proof.* Similar to Equation (9) in the proof of Theorem 2, we have

$$\phi_{(f_T(\epsilon_{L_1},...,\epsilon_{L_n}),\alpha f_T(\epsilon_{L_1},...,\epsilon_{L_n})+f_O(\epsilon_{L_1},...,\epsilon_{L_n})+\epsilon'_O)}((t_1,t_2))\phi_{(\epsilon_T,\alpha\epsilon_T+\epsilon''_O)}((t_1,t_2))$$

$$=\phi_{(f_{\tilde{T}}(\epsilon_{\tilde{L}_1},...,\epsilon_{\tilde{L}_n}),\tilde{\alpha} f_{\tilde{T}}(\epsilon_{\tilde{L}_1},...,\epsilon_{\tilde{L}_n})+f_{\tilde{O}}(\epsilon_{\tilde{L}_1},...,\epsilon_{\tilde{L}_n})+\epsilon'_{\tilde{O}})}((t_1,t_2))\phi_{(\epsilon_{\tilde{T}},\tilde{\alpha}\epsilon_{\tilde{T}}+\epsilon''_{\tilde{O}})}((t_1,t_2)).$$

Because of Condition 6, we have

$$\phi_{(\epsilon_T,\alpha\epsilon_T+\epsilon''_O)}((t_1,t_2)) = \phi_{(\epsilon_{\tilde{T}},\tilde{\alpha}\epsilon_{\tilde{T}}+\epsilon''_{\tilde{O}})}((t_1,t_2)).$$

The subsequent proof proceeds analogously to the proof of Theorem 2. □

**Remark.** A typical scenario where Condition 6 holds is when $f_T$ and $f_O$ are both linear functions and Condition 2 is satisfied. In addition, Condition 6 also holds when $\epsilon_T$ is Gaussian, $\epsilon_{L_1}, ..., \epsilon_{L_n}$ all have a bounded support, and $f_T$ and $f_O$ are arbitrary polynomial functions. In this case, $f_T(\epsilon_{L_1}, ..., \epsilon_{L_n})$ and $f_O(\epsilon_{L_1}, ..., \epsilon_{L_n})$ both have a bounded support, so the condition in this corollary holds.

