# OpenReview forum: "Causal Effect Identifiability in the Presence of Latent Confounders Without Auxiliary Variables"
_ICML.cc/2026/Conference — ICML 2026 regular_

### Official Review · Reviewer_SEhj · 2026-02-25

**Soundness:** 4
**Presentation:** 4
**Significance:** 3
**Originality:** 4
**Overall Recommendation:** 5
**Confidence:** 4

**Summary:**

This paper studies the problem of causal effect identification in the presence of unobserved confounding, without auxiliary variables. The identification results rely solely on the joint distribution of the observed treatment and outcome, based on characterizations of the distribution.

**Compliance With Llm Reviewing Policy:**

Affirmed.

**Final Justification:**

The rebuttal has addressed my main concerns and reinforced my prior assessment.

**Key Questions For Authors:**

1. The statement in the introduction, "In the idealized setting where there is no confounder between the treatment and the outcome, the correlation perfectly reflects the causal mechanism, rendering the causal effect trivially identifiable", is inaccurate. Correlation $\neq$ causation, even without confounding, it still requires additional structural assumptions.
2. Can the authors provide some empirical results when Conditions 1 and 2 are not met?
3. Can the authors discuss how or whether the identification results can be applied to real data?

**Limitations:**

I think the authors should elaborate on the limitations; this seems to be lacking in the current paper.

**Strengths And Weaknesses:**

Strength:

- [Originality] The paper provides an interesting identification result of causal effects based only on the observed joint distribution of treatment and outcome, which is new in the literature.

- [Presentation] The paper is very well-written and easy to follow.

- [Soundness] The proofs in this work look sound to me.

Weakness:

- [Significance] I think the biggest weakness is that, even though the conditions are necessary and sufficient for identification, it is not possible to verify whether these conditions hold or not, given real data. In other words, given real data, one cannot know whether the causal effect is identifiable or not, as one cannot know which condition is met.

---

> ### Author Rebuttal · Authors · 2026-03-29
>
> We sincerely appreciate the reviewer’s insightful comments and positive evaluation.
>
> > Q0: weakness
>
> In the following, we address your concerns regarding the testability of Conditions 1-4 and the significance of our work.
>
> First, we would like to clarify the testability of Conditions 1-4. Specifically, given the joint distribution of $(T,O)$,
> 1. If $T$ is Gaussian, then according to Cramér's decomposition theorem, $\epsilon_T, \epsilon_{L_1}, ..., \epsilon_{L_n}$ are all Gaussian, so Condition 4 holds.
> 2. If $T$ is not Gaussian-separable, then none of $\epsilon_T, \epsilon_{L_1}, ..., \epsilon_{L_n}$ is Gaussian-separable, which implies that $\epsilon_T, \epsilon_{L_1}, ..., \epsilon_{L_n}$ are all non-Gaussian, so Condition 3 holds.
> 3. If $T$ is non-Gaussian but Gaussian-separable, it is theoretically impossible to determine whether $\epsilon_T$ itself is Gaussian or non-Gaussian, so we cannot determine which condition holds based solely on the joint distribution of $(T,O)$. In this specific ambiguous regime, we have to rely on problem-specific domain knowledge.
>
> Second, we wish to highlight that the significance of our contributions is primarily conceptual and theoretical rather than empirical. Specifically, we provide a nuanced and complete characterization of causal effect identifiability without relying on any auxiliary variables, thereby offering a novel perspective on this fundamental problem.
>
>
> > Q1: a statement in the Introduction
>
> We thank the reviewer for pointing out the imprecision in our original phrasing. We will revise the corresponding sentence in the Introduction as follows: In the idealized setting where the causal direction is from the treatment to the outcome and there is no selection bias, the absence of confounders ensures that the observational distribution coincides with the interventional distribution, rendering the causal effect trivially identifiable.
>
> > Q2: empirical results when Conditions 1 and 2 do not hold.
>
> Since our Conditions 1-4 are mutually exclusive and collectively exhaustive, any violation of Conditions 1 and 2 necessarily implies that either Condition 3 or Condition 4 must hold. In this case, the causal effect is unidentifiable, meaning that consistent estimation is impossible even with an arbitrarily large sample size.
>
> For $\mathcal{M}_1$-$\mathcal{M}_6$ from Table 1, we add a noise whose distribution is identical to that of $\epsilon_T$ into $\epsilon_L$. This operation makes $\mathcal{M}_1$-$\mathcal{M}_3$ severely violate Condition 1 but satisfy Condition 3. Also, it makes $\mathcal{M}_4$-$\mathcal{M}_6$ severely violate Condition 2 but satisfy Condition 4. We apply our original estimation procedures to the corrupted $\mathcal{M}_1$-$\mathcal{M}_6$ and summarize the estimation errors below. As expected, the estimation errors remain at high levels.
>
> |sample size|0.5k|5k|50k|500k|
> |-|-|-|-|-|
> |corrupted $\mathcal{M}_1$|0.71|0.62|0.57|0.57|
> |corrupted $\mathcal{M}_2$|0.61|0.57|0.56|0.58|
> |corrupted $\mathcal{M}_3$|0.73|0.58|0.56|0.58|
> |corrupted $\mathcal{M}_4$|0.88|0.76|0.72|0.69|
> |corrupted $\mathcal{M}_5$|0.79|0.67|0.64|0.65|
> |corrupted $\mathcal{M}_6$|0.77|0.71|0.67|0.69|
>
>
> > Q3: application to real data
>
> Applying our results to real data involves a two-step workflow. First, we determine which of Conditions 1-4 holds by leveraging either prior domain knowledge or the distributional tests described in our response to your Q0. Second, if the causal effect is identifiable, it can then be recovered using the estimation procedures detailed in Appendix B.
>
> Finally, we would like to clarify that while we provide estimation procedures in our paper, our primary focus, as explicitly stated in our title, is causal effect identifiability. In other words, our primary contribution lies in solving the fundamental theoretical problem of whether a causal effect can be uniquely determined from the observational distribution, rather than the empirical task of estimating a causal effect given a finite sample.
>
> > Q4: limitations
>
> Thanks for your suggestions. We will discuss the limitations in our revised manuscript. Please see our response to Reviewer `STzf`'s Q3 for more details.

---

> > ### Author Rebuttal · Reviewer_SEhj · 2026-04-02
> >
> > I appreciate the authors response and the additional empirical results.
> >
> > Regarding Q0: what I mean is in how does one use a statistical test to determine which conditions hold. Tests for whether distribution is Gaussian or not given finite sample is statistically challenging: classical tests are known to have low power, and failing to reject does not mean the data is Gaussian.
> >
> > Nevertheless, I keep my score for their theoretical contribution. I still suggest the authors to add a discussion on this point in the final version.

---

> > > ### Author Response · Authors · 2026-04-02
> > >
> > > Dear Reviewer `SEhj`,
> > >
> > > We first sincerely thank you for recognizing our theoretical contribution in characterizing causal effect identifiability without auxiliary variables, which is indeed the primary focus of our paper.
> > >
> > > While our previous response also demonstrated that verifying Conditions 1-4 is sometimes theoretically feasible at the population distribution level, we completely agree with your insightful observation that doing so via statistical tests in a finite-sample regime is challenging. We deeply appreciate your rigorous perspective, and we will add a dedicated discussion on this point into our revised manuscript.
> > >
> > > Thank you again for your time, constructive feedback, and for helping us improve the paper.
> > >
> > > Best regards,\
> > > Authors

---

### Official Review · Reviewer_jdwf · 2026-03-05

**Soundness:** 4
**Presentation:** 4
**Significance:** 2
**Originality:** 3
**Overall Recommendation:** 5
**Confidence:** 4

**Summary:**

This paper presents a sound and complete characterization of identifiability of causal effects of a treatment on an outcome in a linear SCM where there exists $n$ ($n$ is unknown) unmeasured confounders between the treatment and the outcome and no auxiliary variables.
The identifiability conditions are mainly based on the different in Gaussianity between the exogenous noise of the treatment and the those of the $n$ unmeasured confounding.

**Compliance With Llm Reviewing Policy:**

Affirmed.

**Final Justification:**

My concerns have been addressed so I'm more confident about the paper.

**Key Questions For Authors:**

1- Suppose there is a set of measured confounders Z in addition to the set of unmeasured confounders L. It seems to me that the results of this paper can be extended to this case. Of course if the exogenous noise of Z follows the conditions as L then Z and be treated as L. But suppose now that we are in a case where L satisfies condition 1 or 2 and Z satisfied 3 or 4. Intuitively, It seems to me that the confounding bias of Z can be removed by classical methods (adjustment) and the causal effect can be estimated while removing the confounding bias of L using cumulants. Do you agree? If yes, do you have any idea how to combine cumulants with adjustment? If not, can you explain why?

2- For me, the only significant unclear part of the main paper, is the estimation part (for this I had to look to supplementary materials). Can you please move the explanation to the main paper (if the paper is accepted), or at least give a better brief intuition on how the estimation works.  In the estimation of in the case of M1-M3, k is set to 4, why?

3- I appreciate the empirical validity of the results for condition 1 and 2. But I wonder isn't possible to also have results illustrating the results of condition 3 and 4? I expect that in these cases the error should not always monotonically strictly decrease.


4- I understand that the authors did not include baseline methods in the empirical analysis because existing approaches require auxiliary variables. However, it seems that including such methods (together with the required auxiliary variables) could be interesting for the readers. In particular, it would allow readers to compare the coverage of the estimates obtained when identification relies on auxiliary variables versus when no auxiliary variables are used.
That said, I do not believe that including this additional experiment would affect the contribution or impact of the paper. I therefore leave it to the authors to decide whether such an experiment would be useful for readers. I also want to emphasize that choosing not to follow this minor suggestion would not influence my evaluation of the paper.

**Limitations:**

Yes

**Strengths And Weaknesses:**

The paper appears to be sound. I did not check the proofs in supplementary materials but judging on the proofs and intuition available in the main paper, it seems to me that the results are sound. In addition, the empirical results supports the theoretical claims.

The paper is very clear and I have to say that I really appreciated the gray boxes in the paper (for remarks and intuition).

This work focus on a very important and known problem in causal inference which is identifying causal effects. In particular, It focuses on a very special case where: the the generative process is linear, there is unmeasured confounders between the treatment and the outcome, and no auxiliary variables (e.g., no mediator between, the cause and the effect). Therefore, I'm not sure about the significance (complete identification in more general settings were already solved). That said, I do think that the results given by this paper are fairly original and that they are very interesting to the community.

In summary, the strengths of this paper are the results (they are sound and complete) and the presentation. The weakness is the impact of the results in real world applications: 1) it is rare to have a linear SCM (I'm not trying to say that linear models should not be treated and I already checked all the references that authors cited that used linear models. I'm just trying to say that it less impactful than non-parametric results) and 2) it is hard to make assumptions about the noise of the unmeasured confounders (the identifiability results depend on such assumptions).

---

> ### Author Rebuttal · Authors · 2026-03-29
>
> We sincerely thank the reviewer for their constructive comments and high evaluation of our work.
>
> > Q0: weakness
>
> In the following, we separately address your concerns regarding linear SCMs and Conditions 1-4.
>
> 1. (linear SCMs) First, many real-world systems can be approximated by linear SCMs. Second, our insights established in linear SCMs can also be extended to nonlinear scenarios. Please see point  2 of our response to Reviewer `uTba`'s Q1 for more details.
>
> 2. (Conditions 1-4) We can determine which one of Conditions 1-4 holds based on the joint distribution of $(T, O)$ or problem-specific domain knowledge. For a more detailed explanation, please see our response to Reviewer `SEhj`'s Q0.
>
> > Q1: observed confounders
>
> We confirm that our results can indeed be extended to scenarios involving both latent and observed confounders. Specifically, when Condition 1 in Theorem 1 or Condition 2 in Theorem 2 holds and none of the observed confounders is a descendant of any latent confounder, we can regress the treatment and outcome on the observed confounders and work with the residuals. By doing so, the problem reduces to the case without observed confounders. Consequently, Theorem 1 or Theorem 2 naturally applies to this reduced case. A similar reduction procedure is detailed at the beginning of the proof of Theorem 2 in [1].
>
> To empirically validate this extension, we add an observed confounder $Z$ (following the same distribution family as $\epsilon_T$, distinct from $\epsilon_L$) to $\mathcal{M}_1$-$\mathcal{M}_6$ from Table 1 and conduct additional experiments. We report the estimation errors for $\mathcal{M}_1$-$\mathcal{M}_6$ below, showing that the performance on models with $Z$ is very similar to the performance on models without $Z$.
>
> |sample size|0.5k|5k|50k|500k|
> |-|-|-|-|-|
> |$\mathcal{M}_1$|0.54|0.19|0.08|0.02|
> |$\mathcal{M}_1$ w/ $Z$|0.54|0.19|0.07|0.03|
> |$\mathcal{M}_2$|0.27|0.07|0.03|0.01|
> |$\mathcal{M}_2$ w/ $Z$|0.26|0.08|0.02|0.01|
> |$\mathcal{M}_3$|0.44|0.13|0.04|0.02|
> |$\mathcal{M}_3$ w/ $Z$|0.46|0.13|0.04|0.02|
> |$\mathcal{M}_4$|0.64|0.53|0.40|0.35|
> |$\mathcal{M}_4$ w/ $Z$|0.63|0.53|0.44|0.33|
> |$\mathcal{M}_5$|0.47|0.39|0.31|0.26|
> |$\mathcal{M}_5$ w/ $Z$|0.46|0.40|0.31|0.26|
> |$\mathcal{M}_6$|0.50|0.42|0.39|0.32|
> |$\mathcal{M}_6$ w/ $Z$|0.49|0.43|0.37|0.32|
>
> > Q2: estimation procedure
>
> We thank the reviewer for the suggestion and will incorporate a clear, intuitive overview of the estimation procedure into the main text. In the following, we explain why we set $k=4$ for $\mathcal{M}_1$-$\mathcal{M}_3$ from Table 1.
>
> According to the proof of Theorem 1, when estimating the causal effect for $\mathcal{M}_ 1$-$\mathcal{M}_ 3$ from Table 1, we need to find an integer $k > 2$ s.t. $\mathrm{cum}_ {k,0}(T,O) \neq 0$. In our implementation, we default to setting $k=4$ for the following reasons.
> 1. The requirement $\mathrm{cum}_ {4,0}(T,O) \neq 0$ is mild, as $\mathrm{cum}_ {4,0}(T,O)=\mathrm{cum}_ 4(\epsilon_T)$ and the fourth-order cumulant of a non-Gaussian distribution is typically non-zero unless the distribution is specifically designed to be pathological. In contrast, setting $k=3$ might be problematic because $\mathrm{cum}_ {3,0}(T,O)=\mathrm{cum}_ 3(\epsilon_T)$ and any symmetric distribution (e.g., uniform distribution and Laplace distribution) has zero odd-order cumulants.
> 2. Also, $\mathrm{cum}_ {4,0}(T,O) \neq 0$ is testable. Specifically, we can set the null hypothesis to $\mathrm{cum}_ {4,0}(T,O) = 0$ and evaluate it through a hypothesis test such as a bootstrap-based test. If the null hypothesis is rejected, we proceed with $k=4$; otherwise, we simply move to investigate higher-order cumulants.
>
> > Q3: empirical results for Conditions 3 & 4
>
> When Condition 3 or 4 holds, the causal effect is unidentifiable, meaning that consistent estimation is impossible even with an arbitrarily large sample size. The experimental results when Condition 3 or 4 holds are summarized in our response to Reviewer `SEhj`'s Q2. As expected, the estimation errors do not decrease monotonically as the sample size increases.
>
> > Q4: baselines
>
> We agree that contrasting our work with existing methods that rely on auxiliary variables will provide valuable empirical context for the readers. Due to the 5000-character limit of the rebuttal, we will incorporate these additional comparative experiments directly into the appendix of the revised manuscript.
>
> ### Reference
>
> [1] A cross-moment approach for causal effect estimation. NeurIPS 2023.

---

> > ### Author Rebuttal · Reviewer_jdwf · 2026-04-02
> >
> > Thank you for your response. I am now even more confident about my score.

---

> > > ### Author Response · Authors · 2026-04-02
> > >
> > > Dear Reviewer `jdwf`,
> > >
> > > We sincerely thank you for reviewing our rebuttal and for your positive feedback. We deeply appreciate your rigorous and constructive evaluation of our work.
> > >
> > > Best regards,\
> > > Authors

---

### Official Review · Reviewer_uTba · 2026-03-06

**Soundness:** 3
**Presentation:** 4
**Significance:** 3
**Originality:** 3
**Overall Recommendation:** 4
**Confidence:** 4

**Summary:**

The paper studies whether the causal effect of a treatment on an outcome is identifiable in a linear SCM with latent confounders when no auxiliary variables are available. By analyzing four conditions, the paper demonstrates that identifiability is fully determined by the Gaussianity structure of the exogenous noises. Empirical evaluations demonstrate some good performance.

**Compliance With Llm Reviewing Policy:**

Affirmed.

**Final Justification:**

The rebuttal addressed my concern.

**Key Questions For Authors:**

1)	Could the authors clarify more about how their identifiability results fundamentally differ from or extend latent-variable LiNGAM frameworks? It would strengthen the novelty and significance of the contribution.
2)	The empirical evaluation is relatively limited, particularly in assessing the robustness of the method under violations of the model assumptions. Since the theoretical results rely on restrictive assumptions (e.g., linear SCM and specific distributional properties of the exogenous noises), it would be helpful if the authors could conduct a sensitivity analysis to evaluate how the results change when these assumptions are violated. Such experiments would help clarify the practical robustness of the approach in real-world data settings.

**Limitations:**

Yes

**Strengths And Weaknesses:**

Strengths:
1) The writing is clear, and the proof steps are well organized and easy to follow.
2) The paper provides four mutually exclusive and collectively exhaustive conditions based on Gaussianity properties of the exogenous noises.
3) The proofs appear mathematically careful and well structured.

Weaknesses:
1) The framework relies on several restrictive assumptions such as independent exogenous noises and linear structural causal models. These assumptions could limit the application of real-world problem.
2) This paper is not particularly novel in terms of identifiability. The core identification strategy relies on exploiting non-Gaussianity of exogenous noise variables through higher-order cumulants. This technique has already been extensively studied in the LiNGAM literature and related latent-variable causal discovery frameworks.
3) The empirical evaluation is relatively limited, particularly in assessing the robustness of the method under violations of the model assumptions.

---

> ### Author Rebuttal · Authors · 2026-03-29
>
> We sincerely appreciate the reviewer's professional comments and constructive suggestions.
>
> > Q1: assumptions
>
> We would like to clarify that it is reasonable to assume independent exogenous noises and linear SCMs.
>
> 1. (independent exogenous noises) The mutual independence of exogenous noises is a foundational assumption of the structural causal model (SCM) framework (see Definition 6.2 in [1]). Notably, SCM serves as one of the two foundational pillars of modern causal inference (the other is the potential outcomes framework).
>
> 2. (linear SCMs) First, by employing the linearization techniques around the operating point of a system, **real-world systems across various fields can often be approximated by linear SCMs**, for which the extensive literature cited on line 125 of the right column provides many examples. This also explains why many recent causal inference studies center their analyses on linear SCMs. Second, as discussed in Section 5, while our contributions are established in linear SCMs, **our insights can also be extended to nonlinear scenarios**. We provide some preliminary results that establish identifiability of the causal effect in certain nonlinear scenarios in App. C, leaving a more comprehensive investigation for future research.
>
> > Q2: our work vs. lvLiNGAM
>
> Theorem 3.1 in [2] is the most advanced identifiability result in lvLiNGAM (latent variable LiNGAM) that does not require auxiliary variables. In the following, we detail the distinctions between our results and Theorem 3.1 in [2].
>
> 1. In terms of the theoretical scope, **Theorem 3.1 in [2] is subsumed by our Theorem 3**. For instance, lvLiNGAM requires all $\epsilon_T, \epsilon_O, \epsilon_{L_1},...,\epsilon_{L_n}$ to be non-Gaussian, while Condition 3 in our Theorem 3 only requires that $\epsilon_T$ is non-Gaussian and at least one of $\epsilon_{L_1},...,\epsilon_{L_n}$ is non-Gaussian. It is clear that the former is a special case of the latter. For more details regarding connections and distinctions between them, please refer to the Remark following the proof of Theorem 3. Furthermore, **Theorem 3.1 in [2] has no overlap with our Theorems 1, 2, 4**.
>
> 2. In terms of the proof techniques, **while both the proof of Theorem 3.1 in [2] and Part 2 of the proof of our Theorem 3 utilize high-order cumulants, the specific manners of their utilization are different**. For instance, [2] assumes that every non-Gaussian variable has non-zero cumulants of all orders, enabling them to use cumulants of various orders to construct a polynomial $p_{V,l}(b)$ where the causal effect of interest corresponds directly to one of its multiple roots. Without this assumption, we use $r$-th order cumulants to construct a matrix $A(X,Y;p,q,r)$ whose rank holds a complex relationship with the causal effect of interest (see Lemma 3 for more details). Furthermore, **neither Part 1 of the proof of our Theorem 3 nor the proofs of our Theorems 2 and 4 rely on high-order cumulants**.
>
> In summary, our framework has minimal theoretical overlap with [2]. We hope this comparison helps further clarify our novelty.
>
> > Q3: sensitivity analysis
>
> First, in the original paper, the primary aim of our experiments was to empirically validate our theoretical results, so we focused on demonstrating that the estimation error consistently decreases as the sample size grows when the assumptions strictly hold.
>
> Second, we fully agree that a sensitivity analysis is helpful. To address this, we conduct additional experiments.
> 1. (distributional misspecification) For $\mathcal{M}_1$-$\mathcal{M}_3$ from Table 1, we add a non-Gaussian noise (we set its standard deviation to 20% of that of $\epsilon_T$) into $\epsilon_L$ to make these models marginally violate Condition 1. For $\mathcal{M}_4$-$\mathcal{M}_6$, we add a similar Gaussian noise into $\epsilon_L$ to make these models marginally violate Condition 2.
> 2. (mild nonlinearity) For $\mathcal{M}_1$-$\mathcal{M}_6$, we let $T=L^{1.1}+\epsilon_T$ and $O = \alpha T+\beta L^{0.9}+\epsilon_O$ to introduce mild nonlinearity.
>
> We find that the overall performance does not deteriorate significantly, which confirms that our estimation methods are robust to distributional misspecification and mild non-linearity. Due to the 5000-character limit of the rebuttal, we only report the estimation errors for $\mathcal{M}_2$ and $\mathcal{M}_5$ here.
>
> |sample size|0.5k|5k|50k|500k|
> |-|-|-|-|-|
> |$\mathcal{M}_2$|0.27|0.07|0.03|0.01|
> |$\mathcal{M}_2$ w/ distributional misspecification|0.29|0.08|0.03|0.01|
> |$\mathcal{M}_2$ w/ mild nonlinearity|0.24|0.08|0.06|0.05|
> |$\mathcal{M}_5$|0.47|0.39|0.31|0.26|
> |$\mathcal{M}_5$ w/ distributional misspecification|0.51|0.38|0.35|0.31|
> |$\mathcal{M}_5$ w/ mild nonlinearity|0.45|0.38|0.32|0.25|
>
> ### Reference
>
> [1] Elements of causal inference: foundations and learning algorithms. MIT press, 2017.\
> [2] Causal effect identification in lvLiNGAM from higher-order cumulants. ICML 2025.

---

> > ### Author Rebuttal · Reviewer_uTba · 2026-04-01
> >
> > The response is good. I adjust my score.

---

> > > ### Author Response · Authors · 2026-04-02
> > >
> > > Dear Reviewer `uTba`,
> > >
> > > Thank you very much for your swift response. It has been our pleasure to provide clarification and address your concerns.
> > >
> > > Best regards,\
> > > Authors

---

### Official Review · Reviewer_STzf · 2026-03-10

**Soundness:** 2
**Presentation:** 3
**Significance:** 2
**Originality:** 2
**Overall Recommendation:** 4
**Confidence:** 2

**Summary:**

This paper provides complete conditions for causal effect identifiability under linear SCM setting without proxies.

**Compliance With Llm Reviewing Policy:**

Affirmed.

**Final Justification:**

My concerns are addressed.

**Key Questions For Authors:**

Q. Could you explain the main theoretical challenges? After reviewing the proofs in the Appendix, they appeared somewhat straightforward and largely derived directly from the conditions (though this impression may stem from my unfamiliarity with the topic).

**Limitations:**

The limitations of the work are not explicitly discussed, and it would be helpful if the authors could clarify them.

**Strengths And Weaknesses:**

pros: The paper has good readability, and the overall flow is easy to follow.
Cons: I would first like to note that while I am familiar with the algorithmic identification approaches (Tian, Shipitser's), I am not very familiar with the references addressed in this paper.

The paper seems to consider a bow graph setting in which the unobserved confounder between $T$ and $O$ can be viewed as $L_1,\dots,L_n$ in an n-dimensional formulation (since they appear to need to be independent of one another). The scope of the problem addressed in the paper appears to be rather narrow. It is not immediately clear to me what makes the problem particularly challenging.

---

> ### Author Rebuttal · Authors · 2026-03-29
>
> We sincerely thank the reviewer for their time and effort in reviewing our paper.
>
> > Q1: bow graph setting
>
> We would like to clarify that focusing on the bow graph (a causal graph where each latent confounder is a **root** variable with at least two children) does not compromise generality. This is because any complex causal graph (such as one containing causal edges between latent confounders) can be transformed into a bow graph that is observationally and causally equivalent w.r.t. the observed variables. Specifically, Algorithm A in [1] explicitly outlines this conversion process, with Fig. 2 in [1] providing a concrete visual demonstration of a complex causal graph and its equivalent bow graph.
>
> > Q2: theoretical challenges
>
> The main theoretical challenges addressed in our paper primarily stem from the following three distinct aspects.
>
> 1. We investigate causal effect identifiability in a challenging regime where there exist latent confounders but no auxiliary variables. Since most existing studies rely heavily on various auxiliary variables (e.g., mediators, instruments, proxies) to establish identifiability, neither their high-level insights nor low-level techniques can be directly transferred to our setting. This necessitates the development of a different analytical paradigm.
>
> 2. It is challenging to provide a complete and nuanced characterization of identifiability. Previous studies typically stop at establishing sufficient conditions for identification, without systematically discussing whether the causal effect remains identifiable when these conditions are not satisfied. In contrast, we formulate a **novel** set of mutually exclusive and collectively exhaustive conditions, and then carefully select suitable theoretical tools to analyze causal effect identifiability under each condition (e.g., cumulants for Condition 1 and characteristic functions for Condition 2). Moreover, moving beyond merely determining whether the causal effect is identifiable, we also quantify the cardinality of the feasible solution set for the unidentifiable cases.
>
> 3. It is challenging to establish theoretical results without widely-used technical assumptions. For instance, we allow for violations of faithfulness (as mentioned in the Remark following Assumption 1) and we do not assume that each non-Gaussian variable has non-zero cumulants of all orders (this is required in [2]). Without the conveniences provided by these assumptions, we must explicitly account for various pathological cases that are typically ruled out by prior works, thereby increasing analytical complexity in our proofs.
>
> Finally, while the results presented in our paper are easy to follow, we respectfully note that this clarity should not be mistaken for triviality. In fact, achieving this clarity entails iterative refinement, translating intricate technical details into an accessible format.
>
> > Q3: limitations
>
> We will explicitly include the following limitations in our revised paper:
>
> 1. It is not always possible to verify which of Conditions 1-4 holds based solely on the joint distribution of $(T,O)$. We elaborate on this point in our response to Reviewer `SEhj`'s Q0.
>
> 2. Although Appendix C provides some results on causal effect identifiability in nonlinear settings, our exploration of this direction is still at a preliminary stage.
>
> ### Reference
>
> [1] Estimation of causal effects using linear non-Gaussian causal models with hidden variables. IJAR 2008.\
> [2] Causal effect identification in lvLiNGAM from higher-order cumulants. ICML 2025.

---

> > ### Author Rebuttal · Reviewer_STzf · 2026-04-01
> >
> > Thank you for your response. I adjust my score and I will follow up on the discussion with the other reviewers.

---

> > > ### Author Response · Authors · 2026-04-01
> > >
> > > Dear Reviewer `STzf`,
> > >
> > > We would like to express our deep gratitude for your prompt feedback, and we are delighted to have addressed your concerns.
> > >
> > > Best regards,\
> > > Authors

---

### Decision · Program_Chairs · 2026-04-30

**Decision:**

Accept (regular)

**Comment:**

The submission addresses the question of identification of causal effects in the presence of hidden confounders. The contribution is mainly theoretical, with several identifiability/non-identifiability results, but also provides a methodology for estimation with some validation on toy data.

Whiile the reviewers appreciated the clarity of the paper and the theoretical contribution, they also pointed out the limited novelty (in the sense that several related works address this problem in partially overlapping conditions) and quite restrictive assumptions on the distribution of the model that potentially strongly limits its applicability to real world data. The limitation of the experiments to toy synthetic data and the lack of comparison with competitive approaches are also limiting the impact of the contribution. I recommend acceptance if possible due to the theoretical contribution that might be leverage by further work.